

# Variability in methane emissions from West Siberia's shallow boreal lakes

Aleksandr F. Sabrekov[1-3], Benjamin R. K. Runkle[4], Mikhail V. Glagolev[1-3,5,6], Irina E. Terentieva[6,*], Victor M. Stepanenko[7,8], Oleg R. Kotsyurbenko[2,9], Shamil S. Maksyutov[10] and Oleg S. Pokrovsky[1,11]

[1]BIO-GEO-CLIM Laboratory, Tomsk State University, Tomsk, 643050, Russia
[2]UNESCO Department 'Environmental Dynamics and Global Climate Changes', Yugra State University, Khanty-Mansiysk, 628012, Russia
[3]Institute of Forest Science Russian Academy of Sciences, Uspenskoe, 143030, Russia
[4]Department of Biological & Agricultural Engineering, University of Arkansas, Fayetteville, 72701, USA
[5]Faculty of Soil Science, Moscow State University, Moscow, 119992, Russia
[6]Laboratory of Computational Geophysics, Tomsk State University, Tomsk, 643050, Russia
[7]Research Computing Center, Moscow State University, Moscow, 119234, Russia
[8]Faculty of Geography, Moscow State University, Moscow, 119234, Russia
[9]Faculty of Biology, Moscow State University, Moscow, 119992, Russia
[10]Center for Global Environmental Research, National Institute for Environmental Studies, Tsukuba, 305-8506, Japan
[11]Geoscience and Environment Toulouse, Toulouse, 31400 France
[*]previously published as Irina E. Kleptsova

*Correspondence to*: Aleksandr F. Sabrekov (sabrekovaf@gmail.com), Mikhail V. Glagolev (m_glagolev@mail.ru)

**Abstract.** Small lakes represent an important source of atmospheric $CH_4$ from northern wetlands. However, spatio-temporal variations in flux magnitudes and the lack of knowledge about their main environmental controls contribute large uncertainty into the global $CH_4$ budget. In this study, we measured methane fluxes from small lakes using chambers and bubble traps. Field investigations were carried out in July-August 2014 within the West Siberian middle and south taiga zones. The average and median of measured methane chamber fluxes were 0.32 and 0.30 $mgCH_4$ $m^{-2}$ $h^{-1}$ for middle taiga lakes and 8.6 and 4.1 $mgCH_4$ $m^{-2}$ $h^{-1}$ for south taiga lakes, respectively. Pronounced flux variability was found during measurements on individual lakes, between individual lakes and between zones. To analyze these differences and the influences of environmental controls we developed a new dynamic process-based model. It shows good performance with emission rates from the south taiga lakes and poor performance for individual lakes in the middle taiga region. The model shows that, besides well-known controls such as temperature, pH and lake depth, there are significant variations in the maximal methane production potential between these climatic zones. In addition, the model shows that variations of gas-filled pore space in lake sediments are capable to control the total methane emissions from individual lakes. The $CH_4$ emissions exhibited distinct zonal differences not only in absolute values but also in their probability density functions: the middle taiga lake fluxes were best described by a lognormal distribution while the south taiga lakes followed a power law distribution. The latter suggests applicability of self-organized criticality theory for methane emissions from the south taiga zone, which could help to explain the strong variability within individual lakes.

**Keywords**: controls of methane emission, mathematical modeling

## 1 Introduction

Lakes and wetland ponds have strong potential impacts on the methane budget (Travnik et al., 2009) due to their anoxic sediment conditions and often high organic matter content (Zehnder, 1978). This situation is especially true for boreal lakes, covering up to 10% of the zone compared with the extent of only 3% globally (Downing et al., 2006). Lake methane fluxes and their temporal patterns are still poorly constrained, and form a major gap in the northern C budget (Rasilo et al., 2015). Due to its higher global warming potential, methane contributes about 20% of the overall greenhouse effect (Cicerone and Oremland, 1988; Wuebbles and Hayhoe, 2002; IPCC, 2013). Over the past decade, new evidence has demonstrated that





these systems have been underestimated in their contribution to the northern carbon balance (Bastviken et al., 2004; Kortelainen et al., 2004; 2006; Juutinen et al., 2009). Lakes and wetlands ponds can form high CH₄ emission hotspots that contribute largely to landscape-scale CH₄ budgets but create uncertainty for bottom-up regional CH₄ emission estimates due to their small size (Bubier et al., 2005).

Small shallow lakes have high methane emission potential for several reasons. First, methanogenesis is sensitive to temperature conditions (Zeikus and Winfrey, 1976; Dunfield et al., 1993; Hulzen et al., 1999; Kotsyurbenko et al., 2001) and shallow lakes are warmed up quickly during the summer season. Second, small lakes are well mixed so that methane from the sediments can rapidly emit into the atmosphere with minimal oxidation en route (Juutinen et al., 2009). Third, these lakes occupy significant areas in waterlogged regions (Downing et al., 2006) where they receive large inputs of substrate for methanogenesis in the form of terrigenous dissolved organic matter (Segers, 1998).

Methane fluxes from small lakes have great spatial variability (e.g., Casper et al., 2000; Dzyuban, 2002; Kankaala et al., 2004; Bergström et al., 2007), which hampers flux upscaling and modeling of the processes required for making regional and global estimations. This high variability results from multiple environmental controls, including biological (organic matter loading and its mineralization, methane production and oxidation by different groups of microorganisms), physical (temperature, mixing rate, stratification, diffusion and bubble transport rate) and chemical factors (such as concentrations of methane, oxygen and pH) (Rudd and Hamilton, 1978; Bastviken et al., 2004; 2008; Lofton et al., 2013). Due to these variations, the effect of different control factors is complicated and still insufficiently known. In particular we must develop robust relationships between lake CH₄ emissions and their potential controlling factors to facilitate both prediction and spatial extrapolation (Rasilo et al., 2015).

The excess water supply and flat topography with impeded drainage provides favorable conditions for wetland and lake formation in West Siberia. In West Siberian taiga zone wetlands and lakes cover 33% and 4% of the total area respectively (Terentieva et al., 2016). However, there is still a gap of knowledge about methane emission from lakes and, even more rare, their environmental controls. Particularly, we have found only four studies with methane emission data from West Siberian lakes (Gal'chenko et al., 2001; Repo et al., 2007; Glagolev et al., 2011; Sabrekov et al., 2013). These studies indicate that the methane flux from middle taiga lakes is ten times lower in magnitude than from south taiga lakes. An understanding of the nature of this difference is important for modeling and reliably estimating regional methane emission from lakes. In this study, we have the following objectives:

- To estimate methane fluxes from small lakes of the middle and southern taiga;

- To detect key environmental controls of methane emissions on both regional (between zones) and local (within each zone) scale;

- To improve the precision of methane emission modeling.

In order to complete these tasks the following methodology was applied. The best way to take into account the key processes is to construct a process-based model founded on established physical, chemical and biological dependences. Regression procedures, commonly used for identification of environmental controls, cannot fully reflect complicated interactions between different controls, and thus mask a lot of important details. Rather, with process-based modeling, it is possible to test which dependences and which parameters are reliable (at least, for a certain climate zone), and which are not. After taking into account well-known dependences a comparison of predicted and measured values can help to find new potentially important controls.

Since the focus of our work is on lake methane emissions, it is necessary to examine their huge temporal variability and episodic peaks attributed to bubbling, which is challenging to quantify (Walter et al., 2006; Wik et al., 2013). It can be suggested that these stochastic emissions can be modeled using self-organized criticality (SOC) theory (for details on this theory see, e.g., Bak et al., 1988; Bak, 1996; Turcotte, 1999) which can simplify scaling the flux measurements across larger areas. Bubbling is similar to systems showing SOC behavior, where constant external force leads to rapid changes in non-



linear interactions after the reaching of a certain threshold (Bak et al., 1988). Systems with SOC behavior occur in many disciplines, including physics, biology, and economics (Bak et al., 1988). It has been argued that earthquakes, landslides, forest fires, and species extinctions, are examples of SOC in nature (Bak, 1996). To the best of our knowledge, no applications of SOC to bubbling in lakes have been previously published. However, such a behavior of gas bubbles in foam

(Kawasaki and Okuzono, 1996) and in different artificial systems (see, for example, Juodis et al., 2006; Petrashenko et al., 2005) is well-known. Therefore we test whether the measured flux values demonstrate SOC behavior and examine the consequences for upscaling.

## 2 Materials and methods

### 2.1 Study sites

The studied lakes are located in two different boreal zones (both with subarctic climate according to Köppen climate classification) of the Western Siberia Lowland (Russian Federation) (Figure 1). The northern study area is in the middle taiga zone (referred to hereafter as 'MT') about 20–30 km south-west from Khanty-Mansiysk. The southern study area is in the south taiga (referred to hereafter as 'ST') zone about 100-200 km north-west from Tomsk, approximately 900 km

southeast of the MT study area. The climate in both regions is continental with moderate annual rainfall, long and cold winters, and warm summers. Permafrost is absent in both study regions. The main climatic characteristics for the nearest stations (Khanty-Mansiysk for MT and Tomsk for ST) are presented in Table 1.

The MT lakes are mostly acid and humic, have low ionic strength and are surrounded by wetlands (Figure 1; Table 2). Four lakes were selected to cover the range of sediment properties in this zone. Lake Muhrino has peaty sediments with high

mineral content (sandy bedrock), Lakes Babochka and Lebedinoe both have mineral-free peaty sediments and Lake Bondarevskoe has sapropel (flocculated humic material) sediments. The ST lakes are more diverse due to high ground water mineralization. Lakes Bakchar.ryam, Plotnikovo and the three Bakchar.forest Lakes (1–3) represent mesotrophic or eutrophic lakes surrounded by soils rich in clay and grasslands. Lakes Gavrilovka.1 and Gavrilovka.2 represent lakes with low nutrient concentrations and influenced by ground waters with high pH. Lakes Bakchar.bog.1 and Bakchar.bog.2

represent acidic humic wetland lakes with low pH, low ionic strength and low nutrient concentrations. Finally, lake Ob' Floodplain represents floodplain lakes (oxbows) with extremely high nutrient concentrations.

### 2.2 Methods

#### 2.2.1 Field measurements

Field investigations were carried out during summer 2014. We conducted 190 methane flux and 170 carbon dioxide

flux measurements, with 70 and 60 in MT and 120 and 110 in the ST respectively. The total field measurement time varied from 4 to 10 hours per lake, while the average was 6 hours. All measurements were conducted using a rubber boat to prevent any influence on the lake vegetation or sediments. Daytime $CO_2$ and $CH_4$ emissions were measured from the lakes using closed floating chambers. The plexiglas chambers were equipped with plastic bottles to ensure a 5 cm floating depth, and had dimensions of 40×40×30 cm (length×width×height) creating a headspace volume of 0.048 $m^3$. Chambers were covered

by aluminum foil to prevent changes of temperature inside due to solar heating. Four gas samples for both $CH_4$ and $CO_2$ were taken from the fan-mixed chamber headspace at 10–15 min intervals during a chamber closure period of 30–45 min with 12 or 20 ml polypropylene syringes ("SFM", Germany). Prior to sampling, chamber air was used to flush the sampling tube several times. The sample $CH_4$ concentration was measured on a calibrated gas chromatograph "Crystall-5000" ("Chromatec" Co., Ioshkar-Ola, Russia) with a flame-ionization detector (FID) and column (3 m) filled by HayeSep Q (80-

100 mesh) at 70 °C with nitrogen as a carrier gas (flow rate 30 mL $min^{-1}$) or on a calibrated gas chromatograph "KhPM-4"

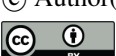


("Hromatograf" Co., Moscow, Russia) with an FID and column (1 m) filled by Sovpol at 40 °C with hydrogen as a carrier gas (flow rate 10 mL min$^{-1}$) within 72 h after sampling. Uncertainty of individual concentration measurement averaged ±0.03 ppm. $CO_2$ concentrations were measured within 4 hours of sampling on an infrared gas analyzer (DX-6100; "RMT Ltd.", Moscow, Russia). The uncertainty of individual $CO_2$ concentration measurements averaged ±2.1 ppm. Fluxes were

calculated from linear regression between chamber headspace $CO_2$ and $CH_4$ concentration versus measurement time using weights inverse to the measurement of gas concentration uncertainty (Kahaner et al., 1989). Fluxes are reported following the sign-convention that exchange from the landscape to the atmosphere are positive.

During the chamber measurements, near-surface water (10 cm depth) was sampled for dissolved $CH_4$. The 20 ml samples were taken with a polypropylene syringe and vigorously shaken for 2 min with a headspace of known $CH_4$

concentration. The headspace $CH_4$ concentrations were then analyzed in a field laboratory on a gas chromatograph as described above. The dissolved $CH_4$ concentrations were calculated with Henry's law accounting for the temperature dependence of solubility (Sander, 2015). Due to logistical problems these measurements were only conducted for three ST lakes (Bakchar.forest.2, Gavrilovka.1 and Plotnikovo).

We tested for potential diurnal variability of stratification and vertical mixing due to the difference between day and

night temperatures that can occur in shallow lakes by examining the lake temperature and oxidation-reduction potential profiles. We found that in this study's lakes these terms do not show strong vertical gradients, and the lakes all belong to the class of "continuous cold polymictic lakes" (Wetzel, 2001). We are therefore confident that our single daytime measurements will not generate a bias in the measured flux estimates (Ford et al., 2002). During flux measurements there were no periods with strong thermal stratification, as temperature gradients between surface and bottom water never

exceeded 2°C. In this concern the studied West Siberian lakes do not correspond to the "summer pattern", described by Ford et al. (2002), when diurnal flux cycles are coupled with afternoon stratification and night mixing, but rather correspond to the "autumn pattern", when there are neither strong gradients nor pronounced diurnal flux variability. A different climate may be the main reason for these differences. In south-eastern Australia, where Ford et al. (2002) obtained their data, summers are hot with a great amount of sunny days when lakes are subjected to substantial heating while the West Siberian taiga climate

is more similar to the Australian autumn. We also neglected the storage flux, because it is important only for stratified lakes, while all study lakes were polymictic (Bastviken et al., 2004).

Submerged funnel gas collectors analogous to those in other measurement campaigns (Huttunen et al., 2001; Repo et al., 2007) were used to monitor $CH_4$ ebullition in the lakes. The gas collectors were 20 and 50 cm diameter funnels feeding into a graduated 500 ml polypropylene cylinder ("Thermo Fisher Scientific", USA), which was fitted with a PVC-tube to a

20 ml polypropylene syringe for sampling. One or two gas collectors were installed randomly near to the study sites in the same day as the chamber measurements for 1–2 days. The net $CH_4$ ebullition was determined from the released gas bubble's volume and its concentration as analyzed on the GC-FID described above.

At each lake site environmental characteristics were measured at three depth levels – 20 cm below water surface, the lake profile mid-point, and 10 cm below sediment depth. At each level, we measured air and water temperatures with

"TERMOCHRON" iButton loggers (DS 1921-1922, DALLAS Semiconductor, USA), pH and oxidation-reduction potential by a "SG-8" ("Mettler Toledo", USA) and electrical conductivity by an "SG-7" ("Mettler Toledo", USA). At the same three depth levels water samples were collected and immediately filtered in pre-washed 30-mL PP Nalgene® flacons through single-use 0.45 µm filter units Minisart (Sartorius, acetate cellulose filter) having a diameter of 25 mm . After discarding the first 20 to 50 mL of filtrate, the filtered solutions for cation analyses were acidified (pH ~ 2) with ultrapure double-distilled

$HNO_3$ and stored in pre-cleaned HDPE bottles. The sample storage bottles were prepared in a clean bench room (ISO A 10,000) and blank samples were used to check the level of pollution induced by sampling and filtration. Major anion ($Cl^-$, $SO_4^{2-}$) concentrations were measured by ion chromatography (HPLC, Dionex ICS 2000) with an uncertainty of 2%. DOC and DIC were analyzed using a Carbon Total Analyzer (Shimadzu TOC VSCN) with uncertainty below 3%. Special



calibration of the instrument for analysis of both form of dissolved carbon in organic-rich, DIC-poor waters was performed as described elsewhere (Prokushkin et al., 2011). Major cations (Ca, Mg, Na, K), Si and trace metal concentrations were measured with an ICP-MS Agilent ce 7500 with In and Re as internal standards and 3 various external standards, measured as check samples between each run of ten lake water samples. Further details about analysis, uncertainties and detection limits are given elsewhere (Pokrovsky et al., 2015; 2016). Sediment layer depth was determined using a peat auger (accuracy is ±0.2 m).

### 2.2.2 Statistical analysis


All statistical analyses were performed using the STATISTICA 8 software ("StatSoft", USA). Ordinary-least square regression ($\alpha = 0.05$) is used to find the significance of the relationship between each environmental variable and the measured $CH_4$ flux (for average and median values across all the individual fluxes for each lake). Stepwise multiple regressions ($\alpha = 0.05$) included parameters such as air temperature (°C), lake depth (LD, m), sediment depth (m), area (ha),

$CO_2$ flux ($mgCO_2$ $m^{-2}$ $h^{-1}$), and on three water depths (surface, middle, bottom): temperatures (°C), pH, oxidation-reduction potential (Eh, mV), electrical conductivity (EC, $\mu S$ $cm^{-1}$), concentrations of DOC, DIC, $SO_4^{2-}$, $Cl^-$, P, Fe, Cu, Ni, Co, and K (mg $l^{-1}$). To estimate the power law distribution parameters $C$ and $\alpha$ in $f(x) = Cx^{-\alpha}$ (where $f(x)$ is a probability density function, and $x$ is a flux) a maximum-likelihood estimate was used (Newman, 2005). The minimum chi-square estimation test was used to check how different probability density functions fit the flux data. In order to test the linearity between flux

rank and absolute value of this flux value in doubly logarithmic coordinates (which is typical for systems having SOC behavior (Bak et al., 1987; 1988; Jensen, 1998; Turcotte, 1999)), the sample of methane fluxes was sorted (where rank 1 indicates the highest magnitude flux). The Levenberg–Marquardt algorithm was used to define coefficients of empirical dependences for methane production on pH and temperature.

### 2.2.3 Model description


To analyze the zonal difference between fluxes, a process-based model reproducing the effect of main environmental controls that are well-known from literature was developed. The model structure is represented in Figure 2, and a full description is given in Appendix A.

The model is constructed similarly to other modern models that have shown good ability to predict methane emissions from lakes and ponds (Stepanenko et al., 2011, 2016; Tan et al., 2015). There are several differences between our

approach and these models. First, in our model, the necessary parameters were each obtained from published literature for the appropriate climate zone (where possible) and averaged across different sources. There was no calibration of model parameters, because we try to test how current scientific knowledge about the methane cycle in boreal lakes can simulate the chamber-measured methane fluxes. In order to avoid using different calibrated constants relating the dependence of methane production from substrates (Stepanenko et al., 2011; Tan et al., 2015) DOC was selected as a single proxy for substrate of

methane emission (Tian et al., 2010). In order to avoid calibrating the strongly variable temperature dependence of methane production and to take into account its potential climatic differences, a climate-sensitive approach was used (see Appendix B). Second, unlike previous models (Stepanenko et al., 2011, 2016; Tan et al., 2015) we have added the influence of pH (Appendix B). Third, gaseous molecular methane diffusion in lake sediments is included in contrast with the previous models (Stepanenko et al., 2011, 2016; Tan et al., 2015). It was introduced because initial numerical experiments

demonstrated that taking into account only liquid $CH_4$ molecular diffusion in the pore space of lake sediments leads to a concentration of dissolved methane in lake water more than an order of magnitude lower than observed in several ST lakes in this study (see Sect. 3 and Table 3) and in other temperate and boreal lakes according to the literature (see Sect. 4.2 for details). Data regarding the gas-filled pore space in shallow lake sediments are very sparse (see Appendix A). Since wetland

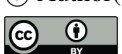



gas-filled pore space occupies from 3% to 18% of total peat volume (Fechner-Levy and Hemond, 1996; Rosenberry et al.,
2003; Strack et al., 2005), we assume that the lake sediment value of this parameter has the same order of magnitude.

This choice of the model framework is based on the data availability, which covers a mix of both spatial and seasonal
variations. Recent models for lake methane emissions (Stepanenko et al., 2011; 2016; Tan et al., 2015) are validated mostly
against seasonal time series taken at singular locations. Thus, it is not clear whether the influence of spatial variability can be
explained according to modern knowledge about environmental controls of methane emission or there are controls which are
valuable on different spatial scales, but not included into models. For example, controls that are relatively stable for a single
lake and on a seasonal scale (climate, lake pH and trophic state, sediment porosity) may not be relevant at greater spatial
scales. Since this paper's obtained flux data cover regional and local spatial variability, we use simple empirical relationships
for controls that are known to be important on these scales: temperature (on a climate-sensitive basis), pH and DOC
concentration (Le Mer and Roger, 2001; Nazaries et al., 2013; Serrano-Silva et al., 2014). The microbial communities of
methanogens and methanotrophs and their dynamics were not simulated (as performed, for example, in Grant and Roulet,
2002; Kettunen, 2003) despite their importance because it is currently not possible to obtain reliable estimates of
microbiological parameters for lakes with different pH and trophic state. Therefore, we compromise between the model's
complexity (which cannot be overly detailed due to the challenge of obtaining reliable data for validation) and data
availability (i.e., that the model should describe the influence of important and measured controls on the scale of the data that
are present).

Input model parameters include the temperature profile of the lake, concentrations of DOC, total phosphorous and the
pH value where these values are measured in the near-bottom water and assumed to be representative for the sediment layer,
lake and sediment depth, latitude and wind speed at the 10 m height. The model outputs are methane and oxygen
concentration profiles, methane ebullition rate and diffusive flux of methane to the atmosphere. The partial differential
equations were solved with MATLAB v. 7.8.0 ("MathWorks", USA). A bootstrap method (Efron and Tibshirani, 1986) was
implemented to find the uncertainty bounds on the modeled fluxes, as follows. First, artificial errors were introduced for
each model parameter using their given standard deviations and a normal distribution. Then, 1000 iterations of these "noisy"
parameter values were used to generate "noisy" flux estimates, and the uncertainty on the predicted flux value was derived as
the standard deviation of these outputs.

**3 Results**

A summary of methane flux measurements is presented in Table 4. The median methane fluxes were 0.3 and 4.1
$mgCH_4\ m^{-2}\ h^{-1}$ for MT and ST lakes, respectively. For MT lakes the median relative uncertainty of the individual
measurements was 10%. For ST lakes the median relative uncertainty of the individual measurements was 20% for fluxes
higher than $1\ mgCH_4\ m^{-2}\ h^{-1}$ and 50% for fluxes lower than $1\ mgCH_4\ m^{-2}\ h^{-1}$. The higher relative uncertainty for smaller flux
observations may be caused by the fact that smaller fluxes are potentially influenced by single rare bubbles. This impact
creates higher scatter in the measured gas concentrations than the relatively constant bubbling driving higher fluxes. The
median fluxes for individual lakes vary from 0.1 to $0.5\ mgCH_4\ m^{-2}\ h^{-1}$ for MT and from 0.8 to $7.4\ mgCH_4\ m^{-2}\ h^{-1}$ for ST.
Methane fluxes in both zones differ significantly (Wilcoxon test, $p < 0.00001$). Average values of pH, EC, P differed
between the two zones with the confidence level of 0.05 in contrast to average DOC, DIC, temperatures, Eh, Fe, and $CO_2$
flux, which were not significantly distinguished. Variability amongst repeated measurements for individual lakes in ST was
higher than in MT: the coefficients of variation were 1.68 and 0.71, respectively. The average dissolved methane
concentration for three ST lakes (Bakchar.forest.3, Gavrilovka.2 and Plotnikovo) is $8.3 \pm 6.3\ mgCH_4\ m^{-3}$ (see Table 3).

Simple regression showed that there were no strong correlations ($R^2 > 0.5$) between environmental variables and
either average or median $CH_4$ flux. The average $CH_4$ flux for all data faintly ($0.5 > R^2 > 0.3$) positively correlates with $CO_2$
flux ($R^2 = 0.43$, $p = 0.012$) and surface [P] ($R^2 = 0.30$, $p = 0.042$). The average $CH_4$ flux for ST lakes correlates faintly





negatively with pH-middle ($R^2 = 0.48$, p = 0.022), pH-bottom ($R^2 = 0.51$, p = 0.020) and lake depth ($R^2 = 0.39$, p = 0.052). Median $CH_4$ flux for all data faintly positively correlates with surface [Cu] ($R^2 = 0.33$, p = 0.030) and $CO_2$ flux ($R^2 = 0.30$, p = 0.043). Median $CH_4$ flux for ST lakes only correlates faintly negatively with bottom [Cu] ($R^2 = 0.44$, p = 0.037). Multiple linear regression by flux median and average did not show any reliable dependences (p $\leq$ 0.05).

The ST lake methane flux is more variable (both for individual lakes and for data combined for all lakes) than the MT lake fluxes (see Table 4). For example, the median coefficient of variation for average flux values from ST lakes (0.87) is more than twice the value from MT lakes (0.36). $CH_4$ flux values in MT and ST lakes are also from different continuous probability density distributions (two-sample Kolmogorov-Smirnov test, p < 0.00001). The combined ST lake flux values correspond to a power law distribution with parameters $C = 0.86 \pm 0.05$ and $\alpha = 1.71 \pm 0.07$ (minimum chi-square estimation

test, p = 0.46, Figure 3a) and do not correspond to a lognormal distribution (p < 0.00001). On the contrary, the fluxes sampled from MT lakes have a lognormal distribution (mean = -1.46 $\pm$ 0.19, variance = 0.72 $\pm$ 0.12, p = 0.25, Figure 3b) and do not have a power law distribution (p < 0.00001). A linear dependence of flux rank on the absolute flux magnitude in doubly logarithmic coordinates (with the exception of a few points at the bounds) is also observed for methane fluxes from ST lakes (Figure 3c), and is not observed for fluxes from MT lakes (Figure 3d). Both power law probability distributions and

linearity in doubly logarithmic coordinates are typical for systems having self-organized criticality (SOC) (Bak et al., 1988). Hence SOC can be used to describe dynamic of methane emission from lakes. In the Sect. 4.3 we try to suggest the features of methane emission processes in lakes that can lead to SOC.

Multiple linear regression did not reveal statistically significant dependences from the environmental factors listed in the Sect. 2.2.2. Therefore, the multiple effect of environmental controls is confounding. Further analysis was provided using

a process-based model (see Sect. 2.2.3 and Appendix A) that reproduced the methane and oxygen production, consumption and transport in lake water and lake sediments.

The modeling results are presented in Figure 4 and in Table 5. The predicted fluxes fit the observed values for ST lakes quite well ($R^2 = 0.76$); however, the model overestimates MT lake fluxes by more than one order of magnitude. Modeled concentrations of dissolved methane in ST lakes vary in a wide range from 0.35 to 21.52 $mgCH_4$ $m^{-3}$ with an

average value 7.63 $mgCH_4$ $m^{-3}$ (recall our observations, where for three ST lakes, the dissolved methane had mean 8.3 $\pm$ 6.3 $mgCH_4$ $m^{-3}$). The modeled fraction of oxidized methane varies from 12% to 40% with an average value 22%. Linear regression analysis of the residual differences between modeled and measured fluxes did not reveal statistically significant dependences from factors listed in the Sect. 2.2.2.

Methane concentrations appear to be strongly underestimated (4–6-fold) for those ST lakes where it was measured

(Table 3). Our numerical experiments showed that the $CH_4$ molecular diffusion in liquids within the pore space of lake sediments by itself could not generate a surface $CH_4$ concentration close to the observed values. These modeled values are very sensitive to the gas-filled porosity of the sediments (Table 3).

Thus, the main differences between observed and predicted methane emissions are that the model:

- overestimated fluxes for MT lakes by more than one order of magnitude;

- underestimated concentration of dissolved methane in both MT and ST lakes (4–6-fold);

Additionally, the data showed extremely high variability of fluxes from ST lakes. Without additional flux monitoring and a greater focus on the driving process mechanisms it may be that this experimental dataset is not suitable for a model comparison or validation effort. In the discussion section we try to suggest where these discrepancies have come from and how they can be explained.

**4 Discussion**

The obtained data indicates that $CH_4$ fluxes are distinctly higher in the ST than MT lakes. They are also in good correspondence with the data reported in these West Siberian zones by other researchers. Fluxes from the MT lakes agree





with Repo et al. (2007) data from near Khanty-Mansiysk sites, where medians for two individual lakes were 0.2 and 1.0 mgCH$_4$ m$^{-2}$ h$^{-1}$. They are also similar to the previously defined flux median of 0.6 mgCH$_4$ m$^{-2}$ h$^{-1}$ (across 51 flux

measurements) for MT wetland lakes and ponds (Glagolev et al., 2011; Sabrekov et al., 2013). The median flux for ST lakes corresponds to the wide interval between 1$^{st}$ and 2$^{nd}$ quartile (4.3 – 23.9 mgCH$_4$ m$^{-2}$ h$^{-1}$, across 82 flux measurements) reported earlier for the wetland lakes and ponds of the same climate zone (Glagolev et al., 2011; Sabrekov et al., 2013).

Comparison with measurements from small (<100 ha) boreal lakes beyond West Siberia is presented in Table 6. Data for MT lakes are in the range of data from other regions in all parameters. Alternatively, methane flux (8.3 versus 1.3

mgCH$_4$ m$^{-2}$ h$^{-1}$ as average sum of diffusive and ebullitive fluxes for lakes from Table 6) and DOC concentration (30 versus 11 mg l$^{-1}$) in ST lakes are considerably higher, but the concentration of dissolved methane is lower (8.3 versus 15.0 mgCH$_4$ m$^{-3}$). A comparison between the model and observational results can produce information on which controls and processes should be measured and examined to improve predictions of boreal lake methane emissions.

**4.1 Differences in methane production between ST and MT lakes**

The significant differences in measured CH$_4$ flux between MT and ST lakes can be explained with the help of the model results. Modeled fluxes from MT lakes are overestimated by one to one and a half orders of magnitude. However, for ST lakes the model meets close to the observations in both the mean level of emissions (where the intercept of the observed vs. predicted flux linear regression is close to zero in comparison with average flux, Figure 4) and the representation of controls (where the slope is close to unity). Hence the question is what model parameter(s) should be changed for MT lakes

to reach good correspondence as well. A first choice could be the potential controlling environmental parameters of CH$_4$ exchange. But pH, DOC, and temperature are in the same range in both zones (although their average values vary) and if the model interprets these values correctly in one zone it is unlikely to shift their interpretation in the other zone. Thus differences in measured emissions between zones could come from parameters that we had modeled as unchanging between zones: the maximal methane production rate (MMPR, Eq. A14 in Appendix A) and/or the maximal intensity of methane

oxidation (Eq. A15). It is doubtful that the key parameters of methane oxidation are different between the zones, because the methanotrophic microbial community is assumed be adaptable to any climate and pH conditions and so the oxidation rate is not heavily influenced by these factors (Le Mer and Roger, 2001; Nazaries et al., 2013; Serrano-Silva et al., 2014). Additionally, methane oxidation is proportional to methane production in lakes where there is no influence of plant methane and oxygen transport (Bastviken et al., 2008; Duc et al., 2010). Therefore, differences in methane flux between zones are

likely caused by differences in MMPR.

We therefore focus on MMPR, which may actually be lower in MT lakes compared to ST lakes due to substrate availability. While the mean difference in DOC between zones is not significant (15 mg l$^{-1}$ versus 30 mg l$^{-1}$ for MT and ST lakes respectively, p = 0.142), if two low-DOC and low emitting Gavrilovka lakes would be excluded from the ST sample, the difference becomes significant (15 mg l$^{-1}$ versus 36 mg l$^{-1}$ for MT and ST lakes respectively, p = 0.028). The DOC

concentration is taken into account in calculations in Michaelis-Menten equation (see Appendix A for details) and it can also influence the MMPR. MMPR implicitly reflects the abundance of methanogenic microbia. Higher substrate availability leads to higher methanogenic biomass and consequently to higher MMPR as simulated explicitly in previous research (Grant and Roulet, 2002; Kettunen, 2003). Greater substrate availability may be caused by higher plant productivity. Methane production correlates positively with plant productivity because root exudates provide additional fresh organic substrates for

methanogens (Whiting and Chanton, 1993; Aulakh et al., 2001; Mitra et al., 2005). NPP in the MT wetlands that surround MT lakes is 40% less than in ST wetlands (Peregon et al., 2008). This mechanism is extremely important for lakes with low level of nutrients where a greater fraction of organic matter is allochthonous. The lower trophic state in MT lakes may also lead to a decrease in the MMPR. It is well known that higher trophic states generate both higher methane production and emission (Bubier, 1995; Segers, 1998; Duc et al., 2010). As a result, a critical concentration for bubble formation is either




not attained at all or is attained at a lower depth in lakes with a lower MMPR. Therefore, a greater fraction of methane diffuses through the water column and is oxidized, further decreasing the flux.

This low-production, low-ebullition hypothesis is supported in this study by measurements with bubble traps (see Table 4): the ebullition flux in MT lakes is less or equal to the diffusion flux calculated as the difference between the flux measured by static chambers and the flux measured by bubble traps. Meanwhile, the ebullition flux in ST lakes is many times higher than the diffusive flux by both model predictions and measurements. Certainly, our field experiments covered a relatively short period and were insufficient for exhaustively estimating methane emission pathways because we lacked time to catch more bubbles. However, data by Repo et al. (2007) obtained in similar lakes with bubble traps are in good correspondence with our measurements.

One can try to estimate the impact of these two possible reasons – "climatic" and "trophic". There are no data about the MMPR in lake sediments in Western Siberia but we can estimate the "climatic" impact driving MMPR differences using data for ST and MT wetlands. According to Kotsyurbenko et al. (2004), the methane production under optimal temperature conditions and without substrate limitation measured in ST wetlands is 110 $mgCH_4$ $m^{-3}$ $h^{-1}$. The same parameter for MT wetlands can be estimated from Kotsyurbenko et al. (2008) as 38 $mgCH_4$ $m^{-3}$ $h^{-1}$. So, taking into account that both sites have similar pH conditions (because both wetlands are acidic, and pH is 4.8 and 4.4 respectively) and trophic state (both wetlands are ombrotrophic *Sphagnum* bogs), the "climatic" methane production in MT wetlands is 3 times lower than in ST wetlands. The "trophic" impact can be estimated using average values of methane production for two groups of Swedish lakes (Duc et al., 2010). These groups were situated within the same region (so, had no "climatic" effect) and approximately correspond in phosphorous concentration to our MT (in (Duc et al., 2010) this group of lakes is labeled "low methane formation", P = 0.011 mg $l^{-1}$) and ST ("high methane formation", P = 0.064 mg $l^{-1}$) lakes. Since phosphorous is known as a main control of methane production in lakes (Bastviken et al., 2004; Juutinen et al., 2009), in the first approximation it represents a difference in the trophic state between groups. For the low methane producing lakes at optimal temperature, methane production is approximately 4 times lower than for the second.

Therefore, the sum of the "climatic" and "trophic" impacts gives a 12-fold reduction for the MMPR value for MT lakes in comparison with ST lakes. If we presume that the model's MMPR value is typical for ST lakes, the MMPR for MT lakes should be 2.60 $mgCH_4$ $m^{-3}$ $h^{-1}$. Model experiments shows that MMPR fitted to the corresponding measured methane fluxes from MT lakes is in the range 1.5–3.7 $mgCH_4$ $m^{-3}$ $h^{-1}$. Hence it can be expected that accounting for both climatic and trophic differences can help to adequately predict MMPR in a variety of lakes and reach a correspondence between measured and modeled fluxes for the MT zone. For these calculations and extrapolations cross-ecosystem comparisons to evaluate the potential effects of both climate and local biogeochemistry on MMPR are needed.

## 4.2 Effect of diffusivity in the lake sediments

Model calculations show that only on average 22% (12-40%) of total produced methane is oxidized (see Table 4). The latter value is lower than the experimentally measured oxidized $CH_4$ fraction from Bastviken et al. (2008) reaching 22–40% for the epilimnion of stratified lakes, similar to this study's lakes because this layer is both oxic and have high turbulence. Both modeled and measured concentrations of dissolved methane in ST lakes are also less than the literature values for different temperate and boreal lakes (Table 6) including the West Siberian MT lakes (Repo et al., 2007); at the same time total methane flux in ST was considerably higher. This situation is in disagreement with the typical pattern where higher methane concentration correlates with greater fluxes (as sum of diffusive and ebullition flux). So, both the oxidized fraction of methane and concentration of dissolved methane seem to be underestimated.

The first possible reason for these differences is that the model has underestimated methane oxidation. Indeed, a comparison of half saturation constants for methane oxidation from different studies showed that this constant for highly producing $CH_4$ samples was greater than for samples with low production rates (Segers, 1998). But the model parameters of



methane oxidation, estimated based on several different sources (see Appendix A), correspond to measured and literature data on concentration of dissolved methane for studied lakes. Model experiments also show that increased oxidation leads to much less concentrations of dissolved methane, moving them even further from the measured values. Methane oxidation can

also vary strongly depending on water chemistry or availability of different metals, such as Ni and Cu which are main micronutrients necessary for enzyme production in both methanogens and methanotrophs (Krüger et al., 2003; Sazinsky and Lippard, 2015). But again the model experiment shows that changing oxidation characteristics cannot increase both the fraction of oxidized methane and concentrations of dissolved methane.

        This pattern could also be explained by the underestimated gas-filled porosity in lake sediments, which is an

important control of dissolved $CH_4$ concentration influencing its diffusion through sediments (see Table 3). Literature data about gas-filled porosity in shallow lake sediments are sparse, while its variability is very high (Valsaraj et al., 1998; Brennwald et al., 2005). However, it is well known that higher silt and mineral content as well as higher bulk density sediments have lower diffusion coefficients for the same values of gas-filled porosity (Clapp and Hornberger, 1978; Moldrup et al., 2003). This mechanism may explain why ebullition was not observed in MT lakes with peat bottoms and higher

diffusion fluxes (Babochka, Lebedinoe) but was observed in lakes with lower organic content in their sediments and lower diffusion fluxes (Muhrino, see Table 4). The same situation was revealed by Repo et al. (2007), who found no ebullition and high diffusive fluxes in lake "MTPond" with peat sediments of several meters. Alternately, the lake "MTlake" of non-wetland origin was characterized by higher mineral content in sediments and lower diffusive flux. As a consequence there was significant ebullition flux in this lake.

The gas-filled porosity influence on methane cycling in lakes could be tested with a quick numerical experiment. Consider doubling the gas-filled porosity for ST lakes to 0.05. This value is still typical for natural shallow lake sediments. For example, according to Valsaraj et al. (1998) the maximal gas-filled porosity is 0.07, a value more than 2 times higher than the 0.025 used by default in our model. In this higher-porosity case the oxidized fraction of produced methane will increase to average 49% (over a wide range from 19-90%). The concentration of dissolved methane will increase to average

27.7 $mgCH_4$ $m^{-3}$, becoming 4 times higher than calculated using the default value of gas-filled porosity. Linearity between the observed and predicted fluxes under this experiment still remains high: Predicted = 1.02·Observed − 1.47 $mgCH_4$ $m^{-2}$ $h^{-1}$, $R^2$ = 0.73. Thus, both the underestimated fraction of oxidized $CH_4$ and concentrations of dissolved $CH_4$ can reach literature and measured values through natural variability in gas-filled porosity.

        It could be concluded that natural variability of gas-filled porosity in the sediments can strongly influence the ratio

between diffusive transport and ebullition and, hence, on the fraction of oxidized methane and total emissions. This variation may result from the extremely non-linear influence of relatively low values of gas-filled porosity on gas diffusivity (Sallam et al., 1984). This non-linearity is related to interconnected water films causing disconnectivity in gas-filled pore space and, thus, reducing gas diffusivity (Moldrup et al., 2003). Unfortunately, data about gas-filled porosity in lake sediments are very sparse and it is difficult to provide a comprehensive analysis of this parameter's influence on methane emission from lakes.

**4.3 Emission uncertainty and self-organized criticality**

        The power-law dynamics of methane emission from ST lakes (Figure 3) are similar with dynamic system behavior in the SOC theory (Bak et al., 1987; 1988; Jensen, 1998; Turcotte, 1999). SOC is based upon the idea that complex behavior can develop spontaneously in certain multicomponent systems whose dynamics vary abruptly. The paper by Bak et al. (1987) contained the hypothesis that systems that i) are driven by some external force and ii) consist of non-linear

interactions amongst their components, may generate a characteristic self-organized behavior. The self-organized state into which systems organize themselves has similar properties as equilibrium systems at their critical point, so they are described as having SOC behavior (Bak et al., 1987). SOC dynamics are assumed to evolve through the contribution of processes at different time scales. The processes driven externally are typically much slower than the internal relaxation processes. A



prototypical example is an earthquake, driven by stress that has slowly accumulated in the earth's crust due to tectonic
activity. This slowly built stress is subsequently released very quickly (in seconds or minutes) in an earthquake (Jensen,
1998). There is an analogous situation in lake sediments, as they become saturated by methane. Methane molecules and
energy input continue much longer and more continuously than the release of bubbles and relaxation to the new steady state
(Scandella et al., 2011).

The separation of relevant time scales is generated by the threshold responses – which build up over time - and
metastability, which awaits a triggering event. In lake sediments, the situation is generated by microorganisms that produce
and emit methane molecules into the surrounding lake water. The methane concentrations increase slowly until a solubility
limit is reached. In this moment a new phase in the form of a bubble is produced. Then the methane concentration inside the
bubble slowly continues to increase until the moment when pressure in the bubble is high enough to do work against forces
preventing its release to the atmosphere (Scandella et al., 2011). When a critical pressure is exceeded, bubbles very quickly
leave sediments via the previously formed channel. The applied force – the buildup of the CH$_4$ concentration - has to
accumulate in order to overcome the critical threshold. This buildup occurs over a much longer time scale than the short time
interval it takes the bubble to be released. The release of accumulated energy is nearly instantaneous in the moment the
bubble moves. If the CH$_4$ molecules were produced very slowly by microorganisms and diffuse in water without ebullition
then no threshold for motion would exist. In this situation the dissolved methane would be continuously released and its
energy dissipated at the same rate as it was produced by the system.

The actual force that the generated bubble of CH$_4$ must overcome depends on the molecular details of how the bubble
interlocks with the sediment particles. As a result, there are a multitude of states where the bubble will remain immobile
even in response to an applied force. When these forces do not pass the release threshold, these states are metastable. The
forces induce strain in the sediment material, corresponding to a certain amount of stored elastic energy. Thus, despite the
bubble-sediment material system existing in an apparently stable, time-independent state, the system is not actually in its
lowest energy state. Rather, the accumulated energy has created one of many metastable states, where a minor increase in
applied force can provoke a number of different responses, including no motion, quick but short motion, or a large jump and
removal of the bubble from the sediment matrix (Scandella et al., 2011; 2016).

There are several practical consequences of SOC behavior of methane emission in lakes. The high values of SD in
Figure 4 show not the low accuracy of measurements but natural spatial and temporal variability of methane emissions from
lakes. Short-term measurements can produce uncertainty if they are extrapolated on a long period or a season. Controls
found to be important from short term measurements may be unreliable on other spatial or temporal scales. Whole season
multiyear measurements in three lakes of Northern Sweden, made by Wik et al. (2013; 2014), confirm this hypothesis. Each
season of their measurements has a unique type of seasonal dynamic with a unique pattern of peaks and falls related to
temperature and atmospheric pressure dynamics (Wik et al., 2013). But the whole season methane budget clearly linearly
correlates with seasonal energy input to lakes (Wik et al., 2014).

Another practical consequence is in the upscaling of flux measurements in lakes for large regions. Once we determine
that the probability distribution law is relevant across all the ST lakes, we can use it for upscaling. The mean value for a
power law distribution is (Newman, 2006):

$$x_{mean} = \left( \frac{C}{2-\alpha} \cdot (x^{-\alpha+2}) \right)\Big|_{x_{min}}^{x_{max}} \tag{1}$$

where $x_{mean}$ is an mean flux value, $x_{min}$ is the minimal flux value, $x_{max}$ is the maximal flux value, and the other parameters
were described in the Sect. 2.2.2. While $x_{min}$, $C$ and $\alpha$ can be easily calculated based on our flux measurement campaign, it is
more complicated to give a reliable estimate of $x_{max}$, because we cannot be sure that in our sample set we have obtained the
maximal possible flux value. In SOC theory $x_{max}$ is infinite, but in real conditions of lakes it is a function of methane
production (as a measure for the applied external force) and sediment diffusivity (as a measure of energy dissipation). The





maximal measured flux in ST lakes in our previous work (Glagolev et al., 2011; Sabrekov et al., 2013) is 359 mgCH$_4$ m$^{-2}$ h$^{-1}$. If we assume this value is the x$_{max,}$ then the mean flux value according to Eq. (1) would be 13.2 mgCH$_4$ m$^{-2}$ h$^{-1}$. This value is 50% higher than the simple average and 220% higher than the median obtained from the flux dataset for ST lakes of the current study. Therefore, use of simple average and median statistics can lead to substantial underestimation of the total methane amount emitted from lakes.

Despite this stochastic behavior of emission, our modeled flux values are in good correspondence with measured fluxes. There are several reasons for this agreement. According to the probability law distribution identified for ST, ten or more flux measurements, as we have performed, allow detection of high flux moments (for example, for our ST flux power-law function the probability to detect a flux with magnitude from 10 to 20 mgCH$_4$ m$^{-2}$ h$^{-1}$ is 0.095) and represent the total emission in the first approximation. The wide range of fluxes and relatively high number of studied lakes also help to obtain good correspondence.

**4.4 Other important controls for regional model development**

Comparison of observed and predicted fluxes can help to reveal other important methane emission controls on a spatial scale. There are two strong site discrepancies for our model: CH$_4$ emission for the Ob' floodplain oxbow lake is strongly underestimated, while the model generates a large overestimation for the Bakchar.forest.1 lake. Both these model-data disagreements can be considered in a context of organic matter quality. In our model it is assumed that organic matter in form of DOC has the same quality for all lakes, but in reality the quality depends on its origin. There are generally two possible sources of this organic matter – plant and algae exudations and decomposition of organic matter (dead plants, different types of peat, gyttja, sapropel etc.); both of them can be autochthonous and allochtonous (Whiting and Chanton, 1993; Cao et al., 1996; Segers, 1998). As a rule, fresh, labile and/or rich in nitrogen organic matter leads to higher methane fluxes (Segers, 1998; Duc et al., 2010).

The Ob' floodplain lake has the highest trophic state (in terms of P concentration) in our sample (see Table 2). So, it is natural to suggest that higher trophic states produce higher MMPR (in this case approximately on 50%) and hence higher emissions for this lake. Phosphorous does not directly influence methane production but strongly positively correlates with concentration of chlorophyll, indicating productivity of algae, and with sediment respiration, indicating higher intensity of organic matter decomposition and higher oxygen consumption by sediments, as reviewed by Pace and Prairie (2005). Higher algae productivity and peat decomposition supply methanogenesis with fresh organic substances, while lower oxygen concentration leads to decreasing of methane oxidation. The Bakchar.forest.1 lake is mesotrophic, but the highest DOC concentration between studied lakes and 2 and more times lower CO$_2$ flux than other lakes with relatively high methane flux (see Table 4). DOC in lake water positively correlates with both plankton and sediments respiration (Pace and Prairie, 2005). It can be suggested that the highest DOC and the lowest CO$_2$ flux for the same lake together demonstrate that this DOC is formed from recalcitrant organic matter leading to lower CH$_4$ production. This hypothesis is in correspondence with findings of Duc et al. (2010) where the similar values of MMPR correlate not with DOC concentration but with the quality of organic matter in form of C:N ratio.

We decided not to compare residuals for the MT lakes because of the small sample size and, as mentioned in Sect. 4.2, possible differences in gas-filled porosity. The latter parameter needs special investigation since now, without further datasets, it requires near arbitrary selection. As CO$_2$ and CH$_4$ fluxes are almost the same for all studied MT lakes, it is interesting to compare them with data of Repo et al. (2007) for the same region. It was obtained that the "MTlake" site corresponds with our dataset's CO$_2$ and CH$_4$ flux values and ebullition-to-diffusive flux ratio. At the same time, much higher diffusive CH$_4$ flux and no ebullition were found for lake "MTPond" of 0.5 ha (see Table 6). The DOC concentration for this lake is not higher than in MT lakes studied by us, but 3-fold higher CO$_2$ flux can indicate better quality of this substrate for methanogenesis, as mentioned earlier in this section. The latter finding can be explained by the fact that "MTPond" is





located in a through-flow poor fen and is partly vegetated (Repo et al., 2007). This setting means that methanogenesis in this lake is supplied by both autochthonous and allochtonous organic matter. In contrast, our study's MT lakes are surrounded by a pine-shrub-sphagnum community that prevents any through-flowing, and are not even partly covered with any vegetation. Therefore, carbon dioxide flux can be useful to predict methane fluxes from shallow lakes, as shown in other studies (Rasilo et al., 2015). Regression dependences of $CO_2$ flux from different controls reliable on a large scale (as obtained, for example, by Kortelainen et al., 2006) can increase the precision of global models of methane emission from lakes.

Another possible important control of $CH_4$ emission is presence of chemical inhibitors. It is well-known that a number of alternative electron acceptors (such as dissolved $NO_3^-$, $Fe^{3+}$, $Mn^{4+}$ and $SO_4^{2-}$) inhibits methane production (Conrad, 1989; Nealson and Saffarini, 1994). But only in one studied lake (the Ob' Floodplain) did the concentration of an inhibitor ($SO_4^{2-}$) in the near bottom water exceed the threshold value 1.92 mg l$^{-1}$ reported in previous research (Kuivila et al., 1989). Despite this fact, emission from this lake is even underestimated by model. The discrepancy can be explained by the fact that an amount of sulfate can prevent methanogenesis only in a small part of sediment layer, as discovered previously by Kuivila et al. (1989). The mechanism of inhibition corresponds with Sabrekov et al. (2016), where two groups of wetlands were distinguished in the forest-steppe zone (next to the south climatic zone after ST) of Western Siberia: one where pore water EC is about 600–800 µS cm$^{-1}$ and $CH_4$ fluxes are 2–4 mgCH$_4$ m$^{-2}$ h$^{-1}$, and one where EC is more than 2000 µS cm$^{-1}$ and fluxes are not higher than 0.1 mgCH$_4$ m$^{-2}$ h$^{-1}$. The maximal EC in the studied lakes is 500 µS cm$^{-1}$. Therefore we can suppose that in the humid climate of the West Siberia taiga zone, there is a small probability of methane inhibition in lakes. This mechanism can be important for regions where evaporation exceeds precipitation, leading to a higher concentration of mineral components in lake water. Certain trace elements can be beneficial for methane production (Basiliko and Yavitt, 2001), but linear regression did not reveal significant correlation between concentrations of these elements in the lake water and model residuals.

## 5 Conclusion

A study of small-size bodies of water in the non-permafrost region of Western Siberia has demonstrated that lake and pond methane fluxes vary on both regional and local spatial scales. Based on the presented model's calculations it can be suggested that it is possible to predict fluxes for individual lakes within the same climate zone with a fair correspondence by taking into account such established controls as temperature, pH and substrate availability. Individual characteristics of lake origin and development, such as sediment gas-filled porosity, trophic state and organic matter quality can also have crucial effects on methane emission.

To successfully predict $CH_4$ fluxes in several zones different values of MMPR should be used. The climate and trophic state may be primary controls of MMPR variability on inter-zonal scale. Searching for simple governing relationships for MMPR on all spatial scales may be the most feasible manner to improve the precision of methane emission modeling.

The constructed ab initio model is much more primitive than more complex recent models (Tan et al., 2015; Stepanenko et al., 2016), but it does not include calibrated parameters, because all parameters can be adopted from the literature as average values from several literature sources for the suitable climate zone. It can be assumed that this approach can be effective for analysis of spatial variability of methane emission which appears to be higher than the temporal (Treat et al., 2007; Olefeldt et al., 2012; Sabrekov et al., 2014). Additionally, controls of spatial variability seem to have lower predictive ability (for example in terms of $R^2$ for regression models) than for temporal variability (Treat et al., 2007; Olefeldt et al., 2012; Wik et al., 2013; 2014; Rasilo et al., 2015).

For global modeling it is important to know, which lakes and with what kind of ecological features and on what season there exists SOC behavior. These lakes can emit significantly more methane because methane bypasses the oxidation filter through ebullition. The most interesting question in this concern is about limits of environmental controls in time and

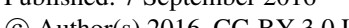



space that define the switch between ebullitive and non-ebullitive regimes. Because of variability of MMPR and diffusivity of lake sediments, the presence of such methane emission "hot spots" as small shallow lakes is expected in any climate zone. However because of their great extent in the taiga and tundra regions, small lakes in those zones are particularly relevant for the global $CH_4$ budget.

550                                         **Acknowledgments**

Support from BIO-GEO-CLIM grant No 14.B25.31.0001, RFBR grants 15-35-50740, 15-05-07622 and 15-44-00091 as well as from the European Union FP7-ENVIRONMENT project PAGE21 under contract no. GA282700 are acknowledged. S. Maksyutov is supported by the Environment Research and Technology Development Fund (2A1202) of the Ministry of the Environment, Japan. We thank all participants of the 2014 summer field campaign. A. Sabrekov thanks J.
Greenwood for his contribution.





### Appendix A. Model description

The model is designed to couple the processes of production, consumption and transport of methane and consumption and transport of oxygen in water column and sediments of shallow boreal lakes. The model structure is similar to other

methane emission models for wetlands (Walter and Heimann, 2000; Tian et al., 2010; Meng et al., 2012) and lakes (Stepanenko et al., 2011, 2016; Tan et al., 2015). The functional forms of process controls were chosen in order to obtain reliable estimates of the governing parameters using publically available information from the appropriate climatic zone. There was no calibration of any model parameters.

Oxygen and methane dynamics in the water column from the water-atmosphere border to the lower boundary of

sediments was modelled using the following equations according to (Tang and Riley, 2014):

$$0 = -\frac{\partial F_{CH_4}(z)}{\partial z} + Prod(z) - Ebul(z) - Ox(z) \tag{A1}$$

$$0 = -\frac{\partial F_{O_2}(z)}{\partial z} - 4 \cdot Ox(z) - Resp(z) \tag{A2}$$

where $F_{CH_4}(z)$ and $F_{O_2}(z)$ (mg m$^{-2}$ h$^{-1}$) are the transport terms, $Prod(z)$, $Ebul(z)$ and $Ox(z)$ (mg m$^{-3}$ h$^{-1}$) are the rates of methane production by methanogens, ebullition and consumption by methanotrophs respectively, $Resp(z)$ (mg m$^{-3}$ h$^{-1}$) is

oxygen consumption by plankton and sediment respiration, $z$ (m) is the spatial coordinate (positive downward) and $t$ (h) is the time. The coefficient 4 reflects the stoichiometric relationship for oxidation, where for each 1 gram of methane, 4 grams of oxygen are necessary according to the equation $CH_4 + 2O_2 = CO_2 + 2H_2O$.

Oxygen and methane diffusion can be written as (Stepanenko et al., 2011; Tan et al., 2015):

$$F_{CH_4}(z) = -D_{CH_4}(z) \cdot \frac{\partial C_{CH_4}}{\partial z} \tag{A3}$$

where $D_{CH_4}(z)$ is the diffusivity for methane and $C_{CH_4}$ (mg m$^{-3}$) is the methane concentration in liquid phase. The resultant diffusion coefficient was calculated as either the sum of molecular diffusivity coefficient within a liquid and eddy diffusivity in the water column or as the sum of molecular transport within both liquid and gas phases in the lake sediment layer (Tang and Riley, 2014):

$$D_{CH_4}(z) = \begin{cases} D_{0,liq,CH_4} \cdot \left(\frac{T(z)}{273} + 1\right)^{1.82} + D_{tur}(z) & \text{if} \quad z \le z_{bot} \\ \left(\Phi(z) - \varepsilon_a(z)\right) \cdot D_{mol,liq,CH_4}(z) + \left(\frac{\varepsilon_a(z)}{\alpha_{CH_4}}\right) \cdot D_{mol,gas,CH_4}(z) & \text{if} \quad z > z_{bot} \end{cases} \tag{A4}$$

where $D_{0,liq,CH_4}$ (m$^2$ h$^{-1}$) is the molecular diffusivity of methane in lake water at 0°C, $T(z)$ (°C) is the water/sediment temperature from field observations, $D_{tur}(z)$, $D_{mol,liq,CH_4}(z)$ and $D_{mol,gas,CH_4}(z)$ (m$^2$ h$^{-1}$) are the wind-induced turbulent diffusivity in water column, molecular diffusivity of methane in sediment pore water and molecular diffusivity of methane through gas-filled pore space of lake sediments, respectively, $\Phi(z)$ (m$^3$ m$^{-3}$) is the total sediment porosity, $\varepsilon_a(z)$ (m$^3$ m$^{-3}$) is the gas-filled porosity, $\alpha_{CH_4}$ (non-dimensional) is the Bunsen solubility coefficient for methane and $z_{bot}$ (m) is the depth to

lake bottom. Molecular diffusion both in liquid and gas phase was taken into account in the sediment layer as it is performed for wetlands (Walter and Heimann, 2000; Zhu et al., 2014). The Penman equation was used for molecular diffusion in the liquid phase, because the fraction of pores in sediments filled with water was high (0.6–0.9), so the equation is quite precise under observed porosity (Moldrup et al., 2000):

$$D_{mol,liq,CH_4}(z) = 0.66 \cdot \left(\Phi(z) - \varepsilon_a(z)\right) \cdot D_{0,liq,CH_4} \cdot \left(\frac{T(z)+273}{298}\right)^{1.82} \tag{A5}$$

Because gas-filled porosity in sediments is very low (0.015–0.07 according to (Valsaraj et al., 1998; Brennwald et al., 2005)), the Penman equation is not accurate in these conditions (Moldrup et al., 2000). As a result, we used the Millington–Quirk equation (Jin and Jury, 1996) which generates a diffusion coefficient similar to experimentally measurements conducted under low gas-filled porosity (Salam et al., 1984; Moldrup et al., 2000):



$$D_{mol,gas,CH_4}(z) = D_{0,gas,CH_4} \cdot \left(\frac{\varepsilon_a^{3.3}(z)}{\Phi^2(z)}\right) \cdot \left(\frac{T(z)}{273} + 1\right)^{1.82} \tag{A6}$$

where $D_{0,gas,CH_4}$ (m$^2$ h$^{-1}$) is the molecular diffusivity of methane in air at 0°C. We neglected the solubility effect in consistency with Stepanenko et al. (2011). The depth profile of $\Phi(z)$ was adopted from (Gadzhiev and Kovalev, 1982) and used for all lakes. The same value of $\varepsilon_a(z) = 0.025$ was chosen for all lakes as an average (Valsaraj et al., 1998; Brennwald et al., 2005). Since gas-filled porosity does not vary strongly with depth for the first 1−3 meters of sediments (Brennwald et al., 2005), it was assumed to be depth-independent. $\alpha_{CH_4}$ was calculated as in (Tang et al., 2010):

$\alpha_{CH_4} = K_{H,CH_4}(T(z)) \cdot \frac{T(z)}{12.2}$            (A7)

where $K_{H,CH_4}(T(z))$ (mg m$^{-3}$ atm$^{-1}$) is temperature-dependent Henry's law constant for methane, calculated as (Sander, 2015):

$$K_{H,CH_4}(T(z)) = K_{0,H,CH_4} \cdot \exp\left(B_{CH_4}\left(\frac{1}{T(z)+273} - \frac{1}{298}\right)\right) \tag{A8}$$

where $K_{0,H,CH_4}$ (mg m$^{-3}$ atm$^{-1}$) is Henry's law constant for methane at 25°C, $B_{CH_4}$ (K) is a coefficient of Henry's law constant

temperature dependence for methane. For calculation of turbulent diffusivity in water in non-neutral conditions, the following scheme (Henderson-Sellers, 1985) was used:

$$D_{tur}(z) = 3600 \cdot \frac{k \cdot w_s \cdot z}{Pr \cdot (1 + 37Ri^2)} \cdot \exp(k^* \cdot z) \tag{A9}$$

$$w_s = 0.0012 \cdot U_{10} \tag{A10}$$

$$k^* = 6.6 \cdot \sqrt{\sin(Lat)} \cdot U_{10}^{-1.84} \tag{A11}$$

$Ri = \frac{1}{20} \cdot \left(-1 + \left(1 + \frac{40 \cdot N^2 \cdot k^2 \cdot z^2}{w_s^2 \cdot \exp(-2 \cdot k^* \cdot z)}\right)^{0.5}\right)$            (A12)

All parameters of Eq. (A9-A12) are specified in the original study (Henderson-Sellers, 1985). Input data include latitude ($Lat$, Rad), Brunt–Väisälä frequency ($N$, s$^{-1}$), and wind speed at 10 m height ($U_{10}$, m s$^{-1}$). Calculations were carried out using wind speed data from closest meteorological stations: Khanty-Mansiysk for MT lakes and Bakchar for ST lakes (Russian Federal Service, 2016). Brunt–Väisälä frequency (a measure of lake stratification stability) calculated basing on

experimental temperature data from surface and bottom of the lake:

$$N = \sqrt{-\frac{g}{\rho_w} \cdot \frac{\rho_s(T(0)) - \rho_b(T(z_{bot}))}{z_{bot}}} \tag{A13}$$

where $g$ (m s$^{-2}$) is the acceleration of gravity, $\rho_w$ (kg m$^{-3}$) is the mean water density, $\rho_s(T(0))$ and $\rho_b(T(z_{bot}))$ (kg m$^{-3}$) are the water density on the surface and the bottom of the lake, respectively, as a function of water temperature, $T(0)$ and $T(z_{bot})$ (°C) at the lake surface and bottom, respectively. Water densities at different temperature were taken from (Weast,

1983). The same Eq. as (A3–A13) but with different Henry's law and molecular diffusivity constants were used for calculating oxygen molecular and turbulent diffusion.

     Methane production in lake sediments was taken into account by multiplying maximal methane production rate (MMPR) $V_{prod,max}$ (mg m$^{-3}$ h$^{-1}$), obtained using an average value according to the review of Segers (1998), and the dimensionless empirical functions. These functions are allowed to vary between 0 to 1 for the following factors:

625          - pH ($f_{prod,pH}$) obtained using extensive data given in (Meng et al., 2012) (see Appendix B for details);

        - temperature ($f_{prod,T}$) obtained using set of literature data (see Appendix B for details);

        - substrate availability, $DOC$ (g m$^{-3}$) by Michaelis-Menten kinetics (Tian et al., 2010) with $K_{prod,DOC}$ (g m$^{-3}$) – Michaelis constant for DOC:

$$Prod(z) = V_{prod,max} \cdot f_{prod,pH} \cdot f_{prod,T} \cdot \frac{DOC}{K_{prod,DOC} + DOC} \tag{A14}$$

Methane oxidation within the profile was calculated based on oxygen and methane concentrations (Michaelis-Menten kinetics) and temperature. The maximal intensity of methane oxidation $V_{ox,max}(z)$ (mg m$^{-3}$ h$^{-1}$) was selected using literature



data separately for the water column $V_{wc,ox,max}$ (Striegl and Michmerhuizen, 1998; Utsumi et al., 1998a, 1998b; Bastviken et al., 2008) and sediment layer $V_{sed,ox,max}$ (Rudd and Hamilton 1975; Lidstrom and Somers 1984; Kuivila et al., 1988):

$$Ox(z) = V_{ox,max}(z) \cdot f_{ox,T} \cdot \frac{C_{CH_4}}{K_{ox,CH_4}+C_{CH_4}} \cdot \frac{C_{O_2}}{K_{ox,O_2}+C_{O_2}} \tag{A15}$$

$$V_{ox,max}(z) = \begin{cases} V_{wc,ox,max} & \text{if} & z \leq z_{bot} \\ V_{sed,ox,max} & \text{if} & z > z_{bot} \end{cases} \tag{A16}$$

where $f_{ox,T}$ (non-dimensional) is function of the methane oxidation temperature dependency varying from 0 to 1, $K_{ox,CH_4}$ and $K_{ox,O_2}$ (mg m$^{-3}$) are the Michaelis constants (the methane and oxygen concentrations at which the methane oxidation rate is at half-maximum). The temperature dependence of methane consumption was also derived as a dimensionless coefficient ranging between 0 and 1:

$$f_{ox,T} = \frac{\exp(b_2 \cdot (T(z))^2 + b_1 \cdot T(z) + b_0)}{b_{max}} \tag{A17}$$

where $b_{max}$ (non-dimensional), $b_0$ (non-dimensional), $b_1$ (°C$^{-1}$), $b_2$ (°C$^{-2}$) are the empirical coefficients.

Oxygen is consumed not only by methane oxidation but also by the plankton respiration in the lake water $V_{pl,resp}(z)$ and sediment respiration $V_{sed,resp}(z)$, both in (mg m$^{-3}$ h$^{-1}$):

$$Resp(z) = \begin{cases} V_{pl,resp}(z) & \text{if} & z \leq z_{bot} \\ V_{sed,resp}(z) & \text{if} & z > z_{bot} \end{cases} \tag{A18}$$

Respiration of plankton in lake water is calculated according to (Pace and Prairie, 2005) as a function of concentration of dissolved phosphorous $C_P(z)$ (mg m$^{-3}$):

$$V_{pl,resp}(z) = 10^{-1.27 + 0.81 \cdot \lg(C_P(z))} \tag{A19}$$

This empirical function was derived from a number of sources for a range of temperature from 11 to 22.5 °C ($R^2 = 0.81$), and thus a temperature correction for this dependence is not required. $V_{sed,resp}(z)$ was calculated using Michaelis-Menten
kinetics (Arah and Stephen, 1998; Walter and Heimann, 2000):

$$V_{sed,resp}(z) = \frac{V_{sed,resp,max} \cdot C_{O_2}}{K_{sed,resp}+C_{O_2}} \tag{A20}$$

where $K_{sed,resp}$ (mg m$^{-3}$) is the Michaelis constant for sediment respiration, $V_{sed,resp,max}$ (mg m$^{-3}$ h$^{-1}$) is maximal sediment respiration rate. The temperature dependence of sediment respiration can be presented in the following form (Arah and Stephen, 1998):

$$V_{sed,resp,max} = V_{10} \cdot \exp\left(\frac{\Delta E}{R} \cdot \left(\frac{1}{283} - \frac{1}{273+T(z)}\right)\right) \tag{A21}$$

where $V_{10}$ (mg m$^{-3}$ h$^{-1}$) is maximal sediment respiration rate at 10°C, $\Delta E$ (J mol$^{-1}$) is activation energy of respiration, $R$ (J K$^{-1}$ mol$^{-1}$) is universal gas constant. Since no data about solar radiation are available, photosynthesis is not taken into account. However, numerical tests show that oxygen does not limit methane oxidation in water column.

Ebullition was calculated under the assumption that emitted methane bubbles immediately reach the surface
(Stepanenko et al., 2011, Tan et al., 2015).

$$Ebul(z) = \max\left\{0; c_e \cdot \left(C_{CH_4}(z) - \alpha_e \cdot C_{cr,CH_4}(z)\right)\right\} \tag{A22}$$

where $c_e$ (h$^{-1}$) and $\alpha_e$ (non-dimensional) are empirical parameters, $C_{cr,CH_4}(z)$ (mg m$^{-3}$) is the critical methane concentration of bubble formation that has been estimated according to Stepanenko et al. (2011):

$$C_{cr,CH_4}(z) = \Phi(z) \cdot K_{H,CH_4}(T(z)) \cdot \left(p_a + \rho_w \cdot g \cdot z - C_{N_2}(z)/K_{H,N_2}(T(z))\right) \tag{A23}$$

where $C_{N_2}(z)$ (mg m$^{-3}$) is the nitrogen concentration in the sediments according to depth profile from (Bazhin, 2001), $K_{H,N_2}(T(z))$ (mg m$^{-3}$ atm$^{-1}$) is the temperature-dependent Henry constant for nitrogen, $p_a$ (Pa) is the atmospheric pressure. Methane flux through the bubbles was calculated by integration within the sediment layer:

$$Flux_{ebul} = \int_{z_{bot}}^{z_{sed}} Ebul(z)\, dz \tag{A24}$$





where $z_{sed}$ (m) is the depth of lower bound of sediments. Gas exchange between bubbles and ambient water was neglected

because its relative impact to methane transport is very small for relatively shallow lakes (Stepanenko et al., 2011; Tan et al., 2015).

As the boundary conditions, we specify zero flux for both gases at the lower bound:

$$\left.\frac{\partial C_{CH_4}}{\partial z}\right|_{z=z_{sed}} = 0; \ \left.\frac{\partial C_{O_2}}{\partial z}\right|_{z=z_{sed}} = 0 \tag{A25}$$

At the upper bound, we specified diffusive methane flux calculated according to (Riera et al., 1999; Bastviken et al., 2004;

Rasilo et al., 2015):

$$\left.\frac{\partial C_{CH_4}}{\partial z}\right|_{z=0} = k_{CH_4} \cdot \left(C_{CH_4}(0) - C_{eq,CH_4}\right) \tag{A26}$$

where $k_{CH_4}$ (m h$^{-1}$) is the so-called "piston velocity", an empirical gas exchange coefficient and $C_{eq,CH_4}$ (mg m$^{-3}$) is

concentration of dissolved methane, corresponding to atmosphere concentration of methane by Henry's Law:

$$C_{eq,CH_4} = P_{CH_4,atm} \cdot K_{H,CH_4}(T(0)) \tag{A27}$$

where $P_{CH_4,atm}$ (atm) is the partial pressure of methane in the atmosphere above the lakes, calculated via concentration using

ideal gas law. $k_{CH_4}$ was calculated as follows (Rasilo et al., 2015):

$$k_{CH_4} = 0.01 \cdot k_{600} \cdot \left(\frac{Sh_{CH_4}(T(0))}{600}\right)^n \tag{A28}$$

where $k_{600}$ (cm h$^{-1}$) and $n$ (non-dimensional) are empirical parameters, $Sh_{CH_4}(T(0))$ (non-dimensional) is the temperature-

dependent Schmidt number. $k_{600}$ was calculated according to best fit from (Crusius and Wanninkhof, 2003):

$$k_{600} = \begin{cases} 0.72 \cdot U_{10} & \text{if} & U_{10} < 3.7 \\ 4.33 \cdot U_{10} - 13.3 & \text{if} & U_{10} \geq 3.7 \end{cases} \tag{A29}$$

Temperature sensitivity of Schmidt number was calculated using interpolation of experimental data from (Jähne et al., 1987)

by third-order polynomial function. Parameter $n$ was assumed –2/3 for wind speed < 3.7 m·s$^{-1}$ and –1/2 for wind speed ≥ 3.7

m·s$^{-1}$ (Crusius and Wanninkhof, 2003). Upper boundary condition for oxygen was calculated in the same way.





**Appendix B: pH and temperature effect on methane production**

Since data about the temperature and pH sensitivity of methane production are highly variable (Dunfield et al., 1993; Segers, 1998; Meng et al., 2012) special consideration is required for these important controls.

$f_{prod,pH}$ is a non-dimensional multiplier reflecting the decrease of methane production under real, non-optimal pH conditions. $f_{prod,pH}$ was calculated using data and the functional form from Meng et al. (2012) but with other coefficient

values:

$$f_{prod,pH} = 10^{(a_2 \cdot pH^2 + a_1 \cdot pH + a_0)}/a_{max} \qquad (B1)$$

We preferred not to use the coefficients given in Meng et al. (2012), because they strongly underestimate production in acidic conditions. Data obtained in a number of Russian lakes (Gal'chenko et al., 2001; Kotsyurbenko et al., 2004; Sabrekov et al., 2012) has shown that both production and emission can be very high even when pH is about 4 and lower. We suppose

that this underestimation is caused mainly by lack of $CH_4$ production data for acidic wetlands. The coefficient of determination, $R^2$ for this dependence, given in Meng et al. (2012), is quite low (0.44) due to scatter in data. This scatter can be explained by variations in other methane production controls. In order to avoid both of these problems we have obtained an empirical function of $CH_4$ production's pH dependence using data binned into 0.5 of pH unity intervals (Figure B1). In order to obtain a function varying from 0 to 1, the fitted function was divided by its maximal value. Therefore we have the

term $f_{prod,pH}$ in the form of Eq. (B1). Fitted parameters are given in Table A1.

$f_{prod,T}$ is a non-dimensional multiplier reflecting the decrease of methane production under real, non-optimal temperature conditions. $f_{prod,T}$ was calculated using the empirical function suggested by O'Neill et al. (1972) (presented within Strashkraba and Gnauk, 1985) for describing the effect of temperature change on photosynthesis:

$$f_{prod,T} = \begin{cases} S^X \cdot \exp(X \cdot (1-S)) & \text{if} & T(z) < T_{max} \\ 0 & \text{if} & T(z) \geq T_{max} \end{cases} \qquad (B2)$$

$$S = (T_{max} - T(z))/T_\Delta \qquad (B3)$$

$$T_\Delta = T_{max} - T_{opt} \qquad (B4)$$

$$X = Y^2 \cdot \left(1 + (1 + C_1/Y)^{1/2}\right)^2 / C_2 \qquad (B5)$$

$$Y = \ln(Q_{10}) \cdot T_\Delta \qquad (B6)$$

where $T_{max}$ (°C) is a maximal temperature (when the process rate drops to zero); $C_1$ (°C) and $C_2$ (°C$^2$) are parameters equal to

40°C and 400°C$^2$, respectively in the original O'Neil model (for photosynthesis) and fitted for methanogenesis; $T_{opt}$ (°C) is optimal temperature (i.e., when the process intensity is maximal); $Q_{10}$ (non-dimensional) is a parameter showing how many times the process rate will grow for each 10°C increase of temperature (for low temperatures).

To constrain the $C_1$ and $C_2$ parameters, we used literature data (Svensson, 1984; Moore et al., 1990; Sass et al., 1991; Dunfield et al., 1993; Parashar et al., 1993; Klinger et al., 1994; Kotsyurbenko et al., 2004) about methanogenesis intensity

under different temperature conditions in different climatic zones. $T_{opt}$ was calculated using linear regression from the average number of days per year with average day temperature higher than 10°C ($N_{T>10}$, days):

$$T_{opt} = 0.055 \cdot N_{T>10} + 13.08 \qquad (B7)$$

This empirical function (B7) was found using the studies cited above as well as others (Cicerone and Shetter, 1981; King et al., 1981; Whalen and Reeburgh, 1988; Frolking and Crill, 1994; Best and Jacobs, 1997). The average annual

number of days with average day temperature higher than 10°C had better correlation with $T_{opt}$ ($R^2 = 0.62$, p = 0.002) than other climatic parameters such as average daily air temperatures, average daily air temperature minimums and maximums of January, July or the whole year.

$T_{max}$ was calculated as a function of $T_{opt}$ using linear regression ($R^2 = 0.74$, p < 0.001), carried out with a help of data from a variety of sources (Van den Berg et al., 1976; Zeikus and Winfrey, 1976; Williams and Crawford, 1984; Svensson,

1984; Sass et al., 1991; Miyajima et al., 1997; Kotsyurbenko et al., 2001; 2004):





$$T_{max} = 1.023 \cdot T_{opt} + 15.29 \qquad\qquad (B8)$$

When $C_1 = 590°C$, $C_2 = 1000°C^2$, $Q_{10} = 2$, the empirical functions (B2-B6) fit almost all the experimental data (Figure B2). The only exception (Figure B2c) can be explained by the fact that emission was measured at a site with low water table depth (10 cm below moss surface) and as a consequence the influence of temperature on methane oxidation confounds with methane production. Usually the temperature optimum of methane oxidation is lower than that for methane production (Segers, 1998). Therefore, the left shoulder of experimental data in Fig. B2c lays below the expected, modeled values.



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



**Table 1**. Main climatic characteristics of the studied areas (for period 1979-2007)

| Site | Coordinates of study areas | Number of lakes studied | Annual precipitation, mm | Mean temperature, °C | | |
|------|----------------------------|-------------------------|--------------------------|------|---------|--------|
| | | | | July | January | Annual |
| MT | 61° N, 69 °E | 4 | 530 | 18.4 | -18.9 | -0.8 |
| ST | 57° N, 83 °E | 10 | 568 | 18.7 | -16.9 | 1.0 |

According to data from: All Russian Research Institute of Hydrometeorological Information - World Data Center (www.meteo.ru)





**Table 2**. Basic characteristics of study lakes (mean values of three measured values – bottom, middle, surface)

| № on map | Lake | Date in 2014 | Lake depth m | Area ha | T °C | pH | EC µS cm$^{-1}$ | Eh mV | DOC | Total P mg l$^{-1}$ | SO$_4^{2-}$ |
|---|---|---|---|---|---|---|---|---|---|---|---|
| | | | | MT lakes | | | | | | | |
| 1 | Bondarevskoe | 24 Aug | 3.7 | 39.15 | 18.4 | 5.3 | 10 | 103 | 5 | 0.011 | n.d. |
| 2 | Lebedinoe | 22 Aug | 2.5 | 7.80 | 17.6 | 4.2 | 35 | 165 | 25 | 0.011 | n.d. |
| 3 | Babochka | 21 Aug | 2.2 | 1.77 | 16.7 | 4.4 | 19 | 150 | 18 | 0.009 | n.d. |
| 4 | Muhrino | 20 Aug | 1.5 | 536.4 | 19.0 | 5.4 | 14 | 95 | 11 | 0.006 | n.d. |
| | | | | ST lakes | | | | | | | |
| 5 | Bakchar.ryam | 19 Aug | 1.4 | 0.45 | 21.1 | 8.3 | 79 | 59 | 48 | 0.076 | 0.33 |
| 6 | Bakchar.forest.1 | 28 Jul | 1.6 | 1.84 | 20.1 | 7.3 | 200 | 132 | 56 | 0.038 | 0.07 |
| | Bakchar.forest.2 | 30 Jul | 2.1 | 1.04 | 14.2 | 7.3 | 271 | 132 | 49 | 0.039 | 0.14 |
| | Bakchar.forest.3 | 18 Aug | 2.3 | 0.70 | 14.5 | 7.3 | 283 | 39 | 26 | 0.013 | 0.57 |
| 7 | Gavrilovka.1 | 15 Aug | 4.7 | 0.93 | 20.7 | 8.2 | 217 | 102 | 8 | 0.011 | 0.32 |
| | Gavrilovka.2 | 16 Aug | 2.5 | 0.19 | 18.4 | 8.3 | 261 | 75 | 10 | 0.011 | 0.30 |
| 8 | Bakchar.bog.1 | 6 Aug | 0.9 | 0.05 | 22.9 | 4.5 | 36 | 311 | 38 | 0.016 | 0.17 |
| | Bakchar.bog.2 | 10 Aug | 2.2 | 0.20 | 21.2 | 4.8 | 46 | 270 | 34 | 0.013 | 0.26 |
| 9 | Plotnikovo | 31 Jul | 1.8 | 0.90 | 19.5 | 7.1 | 183 | 95 | 24 | 0.020 | 0.12 |
| 10 | Ob'.Floodplain | 1 Aug | 1.7 | 4.50 | 18.9 | 7.5 | 260 | 102 | 12 | 0.188 | 3.74 |



**Table 3**. Surface dissolved water $CH_4$ concentration (mg$CH_4$ m$^{-3}$) in three ST lakes, both measured and calculated assuming a range of values of gas-filled porosity

| Lake | Surface dissolved water $CH_4$ concentrations (mg$CH_4$ m$^{-3}$) | | | | |
|------|---|---|---|---|---|
| | Calculated when gas-filled porosity (m$^3$ m$^{-3}$) is | | | | Measured |
| | 0 | 0.025[a] | 0.05 | 0.075 | |
| Bakchar.forest.3 | 1.51 | 2.35 | 8.24 | 35.68 | 10.3 |
| Gavrilovka.2 | 0.13 | 0.21 | 0.73 | 2.43 | 1.20 |
| Plotnikovo | 1.98 | 2.98 | 9.42 | 36.16 | 13.4 |

[a] Value used in model by default (see Appendix A for details)



**Table 4**. Summary of field flux observations (empty cells indicate no data)

| Lake | CH$_4$ flux (static chambers) | | Ebullition CH$_4$ flux[b] | Median CO$_2$ flux (static chambers) |
|---|---|---|---|---|
| | Average ± SD (N[a]) | Median | | |
| | mgCH$_4$ m$^{-2}$ h$^{-1}$ | | | mgCO$_2$ m$^{-2}$ h$^{-1}$ |
| *MT lakes* | | | | |
| Bondarevskoe | 0.5 ± 0.2 (24) | 0.5 | 0.1 ± 0.2 | 17.5 |
| Lebedinoe | 0.3 ± 0.1 (16) | 0.3 | Not found | 22.3 |
| Babochka | 0.1 ± 0.03 (14) | 0.1 | Not found | 24.9 |
| Muhrino | 0.2 ± 0.2 (15) | 0.1 | 0.01 | 16.5 |
| *ST lakes* | | | | |
| Bakchar.ryam | 3.2 ± 2.8 (6) | 1.9 | | 10.2 |
| Bakchar.forest.1 | 7.4 ± 12.5 (10) | 3.1 | 6.8 ± 4.3 | 63.2 |
| Bakchar.forest.2 | 2.6 ± 1.2 (12) | 2.2 | | 87.4 |
| Bakchar.forest.3 | 5.1 ± 2.0 (7) | 5.1 | 3.5 ± 3.1 | 141.4 |
| Gavrilovka.1 | 1.5 ± 1.8 (14) | 0.8 | | 10.1 |
| Gavrilovka.2 | 2.7 ± 2.2 (10) | 1.9 | | 14.0 |
| Bakchar.bog.1 | 8.9 ± 9.5 (12) | 4.7 | 5.2 | 116.8 |
| Bakchar.bog.2 | 8.2 ± 14.1 (14) | 4.1 | | 136.8 |
| Plotnikovo | 7.2 ± 2.5 (8) | 7.4 | | 116.4 |
| Ob' Floodplain | 8.8 ± 7.3 (16) | 5.7 | | 302.8 |


[a] Number of individual flux measurements
[b] Measured using bubble traps; if there were two replicate traps, standard deviation is given.





**Table 5**. Summary of modeling results

| Lake | [$CH_4$] (at 1 m depth) | Modeled diffusive $CH_4$ flux | Modeled ebullition $CH_4$ flux | Modeled total $CH_4$ flux | Difference with real $CH_4$ flux[a] | Oxidized fraction |
|---|---|---|---|---|---|---|
| | $mgCH_4\ m^{-3}$ | $mgCH_4 \cdot m^{-2} \cdot h^{-1}$ | | | | % |
| MT lakes | | | | | | |
| Bondarevskoe | 0.55 | 0.01 | 6.25 | 6.26 | 5.74 | 19 |
| Lebedinoe | 0.95 | 0.02 | 7.23 | 7.25 | 6.95 | 18 |
| Babochka | 2.82 | 0.06 | 6.84 | 6.90 | 6.80 | 18 |
| Muhrino | 28.05 | 0.70 | 5.50 | 6.21 | 6.02 | 16 |
| ST lakes | | | | | | |
| Bakchar.ryam | 10.36 | 0.31 | 2.70 | 3.01 | -0.15 | 26 |
| Bakchar.forest.1 | 21.52 | 0.65 | 9.93 | 10.57 | 3.08 | 13 |
| Bakchar.forest.2 | 3.45 | 0.14 | 1.22 | 1.36 | -1.25 | 36 |
| Bakchar.forest.3 | 2.95 | 0.06 | 5.62 | 5.68 | 0.56 | 23 |
| Gavrilovka.1 | 0.35 | 0.01 | 1.63 | 1.64 | 0.15 | 35 |
| Gavrilovka.2 | 0.76 | 0.02 | 0.84 | 0.85 | -1.86 | 40 |
| Bakchar.bog.1 | 15.60 | 0.16 | 9.47 | 9.78 | 0.87 | 12 |
| Bakchar.bog.2 | 4.71 | 0.09 | 7.99 | 8.08 | -0.16 | 22 |
| Plotnikovo | 12.63 | 0.38 | 7.33 | 7.71 | 0.50 | 17 |
| Ob' Floodplain | 10.49 | 0.23 | 5.07 | 5.30 | -3.49 | 20 |

[a]Difference between modelled total flux (as a sum of diffusive and ebullition flux) and measured average flux






**Table 6**. Summary for temperate and boreal lakes with area <100 ha. Empty cells mean no data, storage fluxes are not shown and do not have values more than 1 mgCH₄ m⁻² h⁻¹

| Reference | Coordinates | MAT[a] | MJT[b] | DOC | Total P | Surface [CH₄] | Average CH₄ flux | |
|---|---|---|---|---|---|---|---|---|
| | | | | | | | Diffusive | Ebullition |
| | | °C | | | mg·l⁻¹ | mgCH₄·m⁻³ | mgCH₄·m⁻²·h⁻¹ | |
| Juutinen et al., 2009 | Finland | -2.8 | 12.3 | 5 | 0.006 | 5.1 | 0.06[d,e] | |
| | | – | – | 9 | 0.014 | 6.4 | 0.14[d,e] | |
| | | 5.9 | 17.2 | 17 | 0.024 | 4.0 | 0.12[d,e] | |
| Repo et al., 2007 | 66° N, 75° E | -4.7 | 16.3 | 11 | 0.044 | 4.6 | 0.25 | 0.09 |
| Huttunen et al., 2003 | 63° N, 28° W | 3.4 | 17.5 | | 0.056 | | 0.61 | 1.7 |
| Repo et al., 2007 | 61° N, 70° E | -0.8 | 18.2 | 17 | 0.005 | 41.6 | 1.71 | Not found |
| | | | | 10 | 0.006 | 4.5 | 0.34 | 0.25 |
| Casper et al., 2000 | 59° N, 3° W | 13.0 | 19.9 | | 0.60 | 17.6 | 0.30 | 8.3 |
| Bastviken et al., 2004[c] | 59° N, 15° E | 5.8 | 16.5 | 11 | 0.014 | 10.7 | 0.11 | |
| Rudd and Hamilton, 1978 | 50° N, 95° W | 3.1 | 19.7 | 9 | 0.035 | | 0.91[d] | |
| | | 4.3 | 18.3 | 22 | 0.028 | | 0.15 | 0.18 |
| Bastviken et al., 2008 | 46° N, 90° W | 4.3 | 18.3 | 4 | 0.010 | 30.4 | 0.60 | 0.41 |
| | | 4.3 | 18.3 | 5 | 0.008 | 20.8 | 0.39 | 0.68 |
| Bastviken et al., 2004[c] | 46° N, 90° W | 4.3 | 18.3 | 10 | 0.020 | 17.8 | 0.35 | 0.63 |
| Smith and Lewis, 1992 | 40° N, 106° W | 1.7 | 13.7 | 10 | | 16.3 | 1.1 | |
| This study, MT lakes | 61° N, 69° E | -0.8 | 18.2 | 15 | 0.009 | | 0.32[f] | 0.28 |
| This study, ST lakes | 57° N, 83° E | 1.0 | 18.7 | 30 | 0.043 | 8.3 | 8.60[f] | 4.30 |

[a]Mean annual temperature
[b]Mean July temperature
[c]Average values according to Table 1 data for central Sweden (59° N, 15° E) and Land O'Lakes, Wisconsin, USA (46° N, 90° W) are given
[d]Sum of diffusion and storage flux
[e]Medians are given. There are no average values in original work.
[f]Static chamber values, presenting sum of diffusive and ebullition flux values, are given.





**Table A1**. List of the model parameters.

| Parameter | Description | Value | Units | Reference |
|---|---|---|---|---|
| $a_0$ | Coefficients of the dependency of the methane production on pH, $f_{prod,pH}$ | -3.5172 | | Recalculated data from (Meng et al., 2012) (see Appendix B) |
| $a_1$ | | 1.1217 | $pH^{-1}$ | |
| $a_2$ | | -0.0921 | $pH^{-2}$ | |
| $a_{max}$ | | 0.7905 | | |
| $b_0$ | Coefficients of the dependency of methane oxidation on temperature, $f_{ox,T}$ | -3.6945 | | Glagolev, 2006; $b_{max}$ was chosen so that the function ranges between 0 and 1 |
| $b_1$ | | 0.1486 | $^oC^{-1}$ | |
| $b_2$ | | -0.0029 | $^oC^{-2}$ | |
| $b_{max}$ | | 0.1668 | | |
| $B_{CH_4}$ | Coefficient of Henry's law constant temperature dependence for methane | 1700 | K | Sander, 2015 |
| $B_{N_2}$ | Coefficient of Henry's law constant temperature dependence for nitrogen | 1300 | K | Sander, 2015 |
| $B_{O_2}$ | Coefficient of Henry's law constant temperature dependence for oxygen | 1500 | K | Sander, 2015 |
| $C_1$ | Empirical coefficients for temperature dependence of methane production, $f_{prod,T}$ | 590 | $^oC$ | see Appendix B |
| $C_2$ | | 1000 | $^oC^2$ | |
| $c_e$ | Parameter, defining velocity of bubble formation | 1.008 | $h^{-1}$ | Walter and Heimann 2000 |
| $D_{0,gas,CH_4}$ | Diffusion coefficient for $CH_4$ in the air at 0 $^\circ$C | 0.068 | $m^2\,h^{-1}$ | Arah and Stephen 1998 |
| $D_{0,gas,O_2}$ | Diffusion coefficient for $O_2$ in the air at 0 $^\circ$C | 0.065 | $m^2\,h^{-1}$ | Arah and Stephen 1998 |
| $D_{0,liq,CH_4}$ | Diffusion coefficient for $CH_4$ in the water at 25 °C | $5.4\cdot10^{-6}$ | $m^2\,h^{-1}$ | Arah and Stephen 1998 |
| $D_{0,liq,O_2}$ | Diffusion coefficient for $O_2$ in the water at 25 °C | $8.6\cdot10^{-6}$ | $m^2\,h^{-1}$ | Arah and Stephen 1998 |
| $g$ | Gravitational acceleration | 9.81 | $m\cdot s^{-2}$ | Weast, 1983 |
| $K_{0,H,CH_4}$ | Henry's law constant for methane at 25 °C | 0.021 | $mg\,m^{-3}\,atm^{-1}$ | Sander, 2015 |
| $K_{0,H,N_2}$ | Henry's law constant for nitrogen at 25 °C | 0.017 | $mg\,m^{-3}\,atm^{-1}$ | Sander, 2015 |
| $K_{0,H,O_2}$ | Henry's law constant for oxygen at 25 °C | 0.040 | $mg\,m^{-3}\,atm^{-1}$ | Sander, 2015 |
| $K_{ox,CH_4}$ | Michaelis $CH_4$-constant for methane consumption | $116\pm39$ | $mg\,m^{-3}$ | Rudd and Hamilton 1975; Lidstrom and Somers 1984; Kuivila et al., 1988 |
| $K_{ox,O_2}$ | Michaelis $O_2$-constant for methane consumption | $1019\pm1019$ | $mg\,m^{-3}$ | Bender and Conrad 1994 |
| $K_{prod,DOC}$ | Michaelis DOC-constant for soil methane production | $10\pm7$ | $g\,m^{-3}$ | Lokshina et al., 2001 (review); Tian et al., 2010 |
| $K_{sed,resp}$ | Michaelis constant for sediment respiration | $7040\pm2500$ | $mg\,m^{-3}$ | Frenzel et al., 1990; Arah and Stephen 1998; |
| $P_{CH_4,atm}$ | Atmospheric partial pressure of $CH_4$ | $1.9\cdot10^{-6}$ | atm | measured value |
| $P_{O_2,atm}$ | Atmospheric partial pressure of $O_2$ | 0.2095 | atm | Weast, 1983 |
| $Q_{10}$ | Empirical coefficient for temperature dependence of methane production, $f_{prod,T}$ | 2 | | see Appendix B |
| $R$ | Universal gas constant | 8.314 | $J\cdot K^{-1}\cdot mol^{-1}$ | Weast, 1983 |
| $V_{10}$ | Maximal respiration rate at 10°C | $27000\pm12000$ | $mg\,m^{-3}\,h^{-1}$ | Yavitt et al., 1987; Arah and Stephen 1998; Thamdrap et al., 1998 |
| $V_{prod,max}$ | Maximal rate of methane production | $31.3\pm24.4$ | $mg\,m^{-3}\,h^{-1}$ | Segers, 1998 (review) |
| $V_{sed,ox,max}$ | Maximal rate of $CH_4$ consumption in the sediments | $228\pm153$ | $mg\,m^{-3}\,h^{-1}$ | Rudd and Hamilton 1975; Lidstrom and Somers 1984; Kuivila et al., 1988 |
| $V_{wc,ox,max}$ | Maximal rate of $CH_4$ consumption in the water column | $4\pm2.4$ | $mg\,m^{-3}\,h^{-1}$ | Striegl and Michmerhuizen, 1998; Utsumi et al., 1998a, 1998b; Bastviken et al., 2008 |
| $\alpha_e$ | Coefficient describing concentration when bubble formation starts | 0.4 | | Wania, 2007 |





| | | | | |
|---|---|---|---|---|
| $\varepsilon_a$ | Gas-filled porosity | 0.025 | $m^3\ m^{-3}$ | Valsaraj et al., 1998; Brennwald et al., 2005 |
| $\rho_w$ | Water density | 1000 | $kg\ m^{-3}$ | Weast, 1983 |
| $\Delta E$ | Activation energy of respiration | 50000 | $J\ mol^{-1}$ | Arah and Stephen 1998; Thamdrap et al., 1998; |





**Figure 1**. Studied lakes in MT (left panel; shades of yellow corresponds to floodplain, brown – to forests, green – to pine-shrub-sphagnum communities, purple – to ridge-hollow complexes) and ST (right panel; shades of green corresponds to forests and grasslands, blue – to wetlands) on Landsat satellite images.

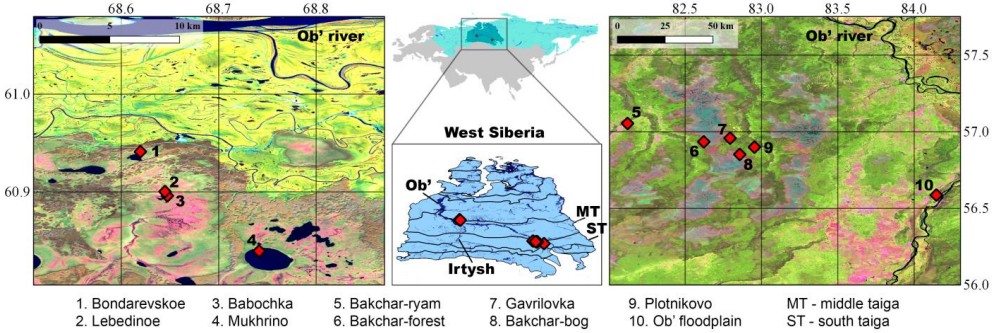

| | | | |
|---|---|---|---|
| 1. Bondarevskoe | 3. Babochka | 5. Bakchar-ryam | 7. Gavrilovka | 9. Plotnikovo | MT - middle taiga |
| 2. Lebedinoe | 4. Mukhrino | 6. Bakchar-forest | 8. Bakchar-bog | 10. Ob' floodplain | ST - south taiga |




**Figure 2**. Schematic representation of the model structure. The one-dimensional column is divided into lake water and sediments. The forcing consists of lake and sediment depths, the water $T_{water}(z)$ and sediment $T_{sed}(z)$ temperature, DOC and phosphorous concentration [P] and pH. $CH_4$ production occurs only in the sediments. The methane production rate $R_{prod}(z)$ is a function of the sediment temperature, the DOC, which is taken as a measure for substrate availability, and pH. $CH_4$ oxidation is calculated in the different way for sediments and for lake water. The $CH_4$ oxidation rate $R_{oxid}(z)$ follows Michaelis-Menten kinetics for both methane and oxygen and is a function of the water and sediment temperature. The water respiration rate WaterResp(z) is a function of the water temperature, the phosphorous concentration, which is taken as a measure for abundance of phytoplankton. The sediment respiration rate SedResp(z) is a function of the sediment temperature. Both water and sediment respiration follows Michaelis-Menten kinetics for oxygen uptake. Transport of $CH_4$ in lake sediments proceeds by (1) molecular diffusion through gas-filled and water-filled pores, (2) ebullition, which is the formation of gas bubbles in the sediment layer and their immediate ascent to the water surface. Transport of $CH_4$ in lake water proceeds by (1) wind-induced turbulent diffusion (2) molecular diffusion in water. Transport of $O_2$ is the same except ebullition. The model calculates methane fluxes to the atmosphere and $CH_4$ and $O_2$ concentration profiles in the lake water and sediments.

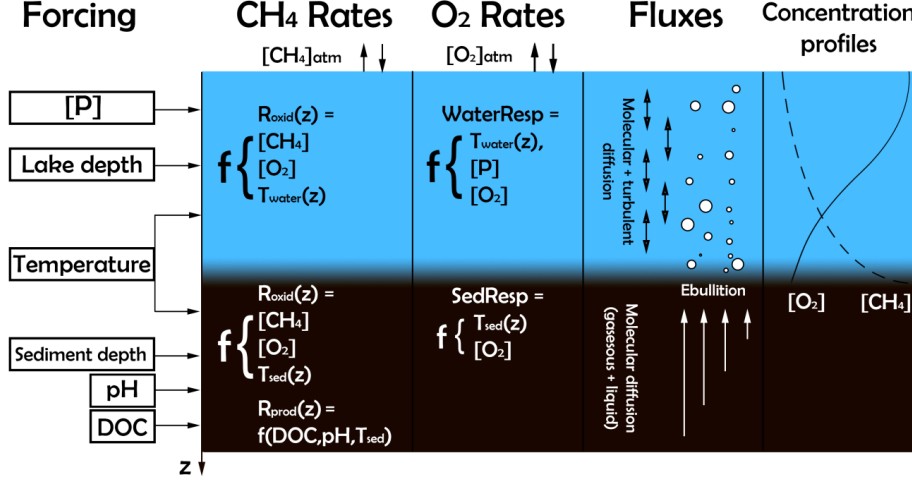



**Figure 3**. Flux data for (a) ST and (b) MT lakes in log-log scale and probability distribution fitting: (c) power law for ST lakes fluxes and (d) log-normal for MT lakes fluxes. Rank 1 indicates the highest magnitude flux. Note the strong difference in the y-axis scaling between the two regions.

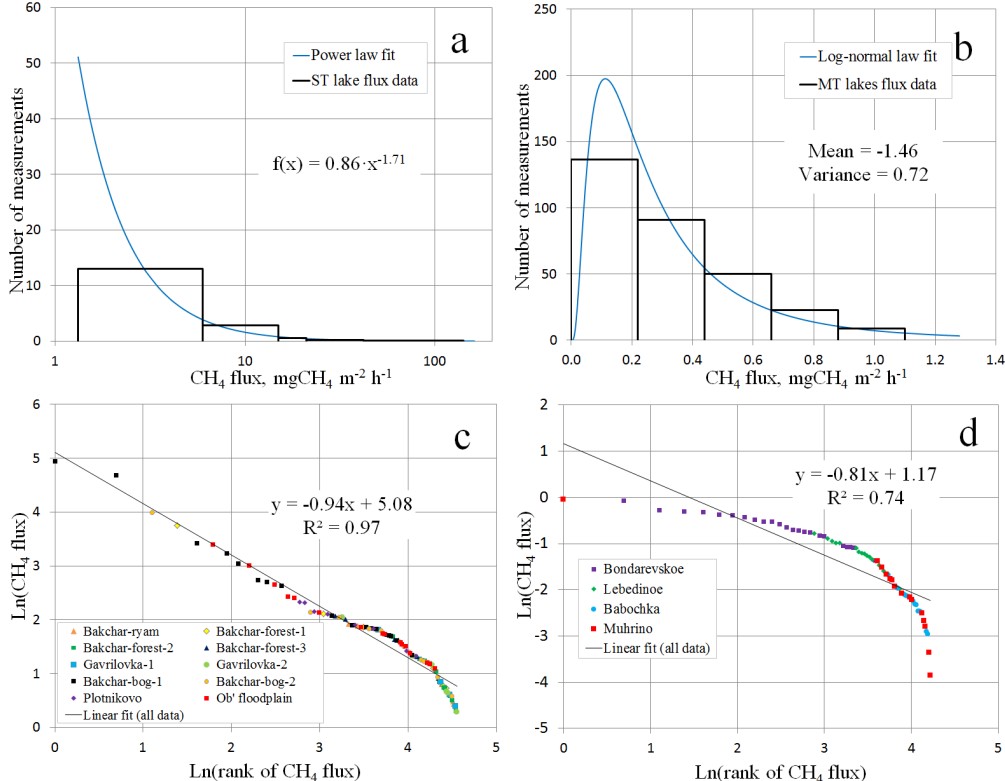





**Figure 4**. Observed versus predicted values of methane flux. Whiskers denote ±1SD. Predicted flux uncertainties are calculated from bootstrapping (see Sect. 2.2.3) and are explained by high uncertainty in model parameters adopted from literature (see Appendix A). High magnitudes of observation SD's may be explained by ebullition and SOC behavior of lakes as methane sources (see Sect. 4.3). Note that predicted flux uncertainties are higher than the magnitude of observation SD's for the MT lakes but less for the ST lakes.

1105

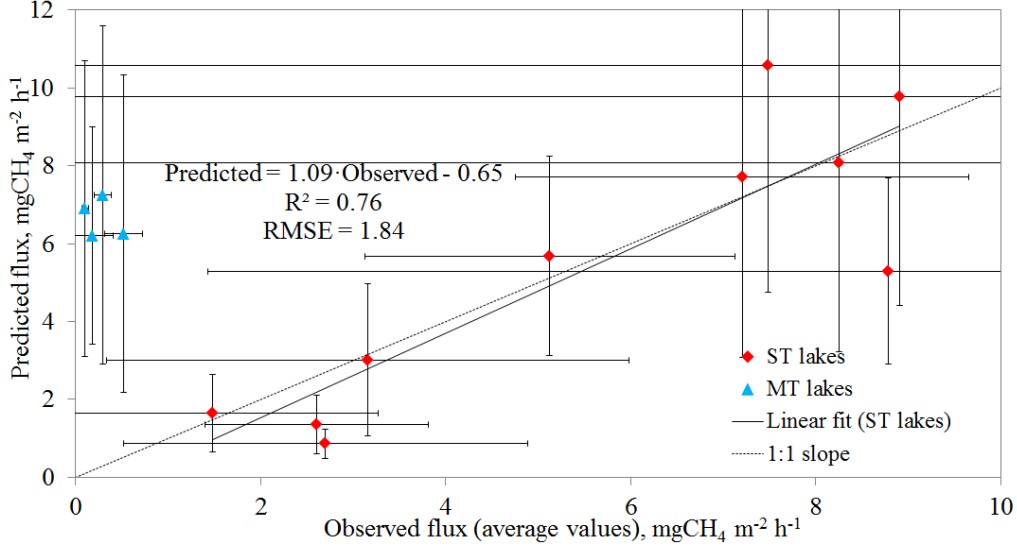





**Figure B1**. Empirical function of relative methane production dependence on pH. The function from Meng et al. (2012) is given for comparison. Whiskers denote ±1SD for binned values of production, according to data presented in Meng et al. (2012).

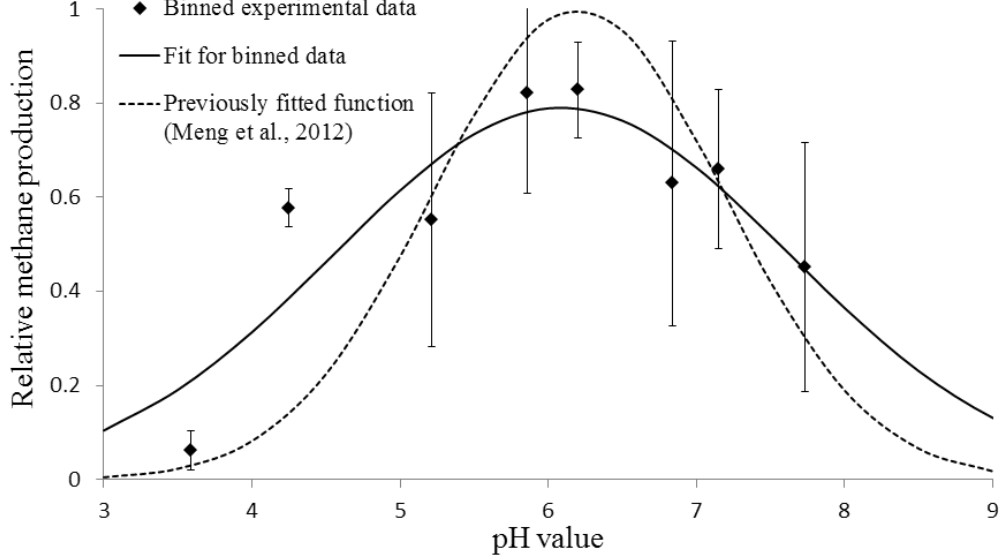

1110



**Figure B2**. Empirical function of relative methane emission (a-c) and production (d-h) dependence on temperature. Black squares are experimental data, solid lines represent the fitted empirical function using Eq. (B2-B8). Standard deviations are given as whiskers for investigations where they have been presented.

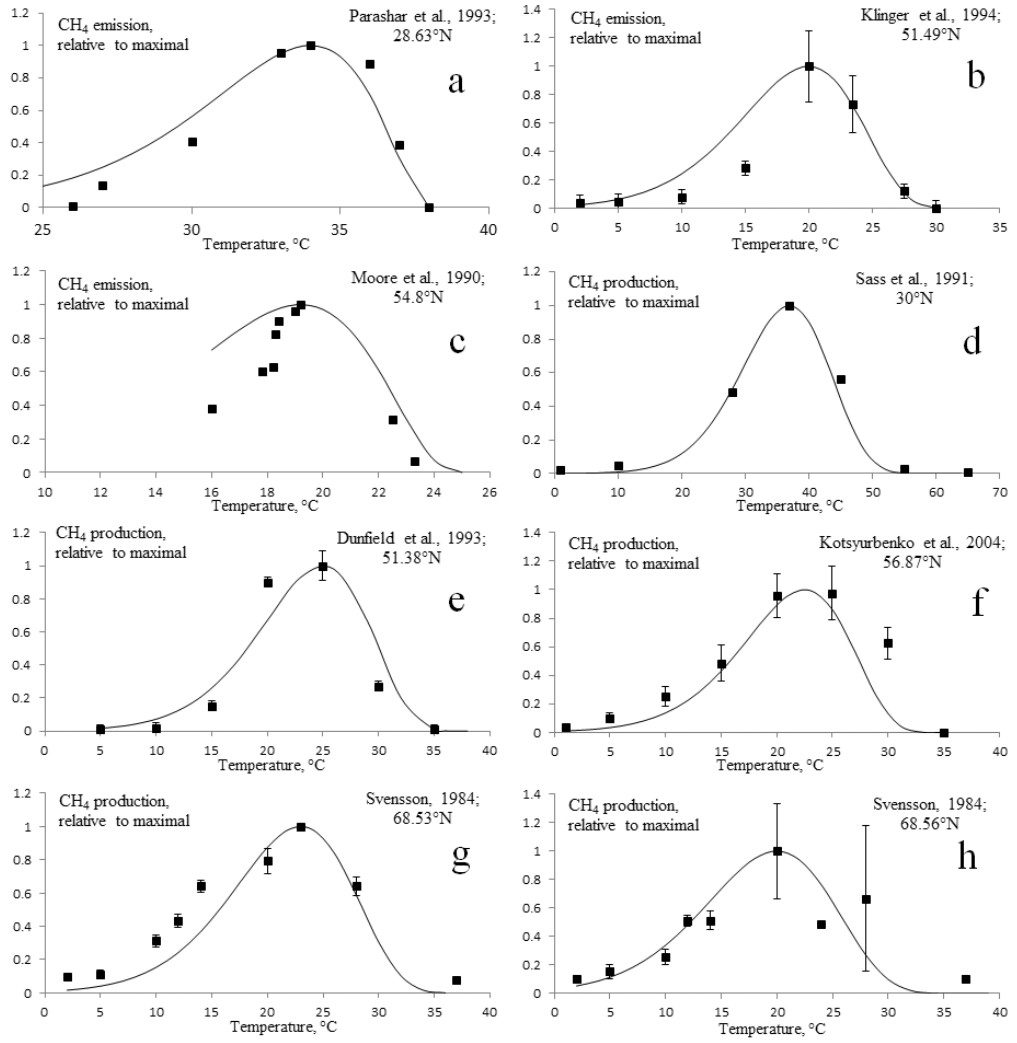

1115