# Peer review of "Variability in methane emissions from West Siberia's shallow boreal lakes on a regional scale and its environmental controls"

_Biogeosciences, 2016_

## Referee Comment (RC1) · Anonymous Referee #1 · 14 Oct 2016

General comments

This manuscript presents spatio-temporal variation in summertime methane fluxes from small lakes within the West Siberian middle and south taiga zones. Pronounced flux variability was found between individual lakes and between zones. The methane data together with 25 environmental controls was used to develop a new dynamic process-based model. The model showed good performance with emission rates from the south taiga lakes and poor performance for individual lakes in the middle taiga region suggesting that besides the well-known controls such as temperature, pH and lake depth, there are significant variations in the maximal methane production potential between these climatic zones.

I think this manuscript is an important addition to the understanding of the methane dynamics and sources globally, because small lakes are one important methane source. Small lakes are numerous in the vast Siberian taiga, and little is known about their methane emissions. A good model predicting the methane emissions will be a powerful tool to extrapolate the predictions to the larger areas. The manuscript requires some revisions and technical corrections to make it easier to read and understand.

The paper addresses relevant scientific questions and presents tools (a model) to evaluate the methane release from the West Siberian lakes. The results are an important step towards the exhaustive model of the methane release from lakes, which would greatly improve the possibilities to create global scenarios of the methane release from the aquatic systems. The scientific methods and experiments are valid and mainly clearly outlined, but the authors could improve the presentation of the methods in the manuscript in order to make it easier to read and understand. The description of the calculations concerning the model are complete and precise. In the paper, the authors refer properly to the earlier methane studies. The title reflects the contents of the paper and the abstract provides a concise and complete summary of the work. The overall presentation needs a few specifications and maybe some structural changes. I also recommend the language check by the native English speaker. The usage of abbreviations needs to be checked and revised. The supplementary material is appropriate and useful and gives support to the findings in the manuscript.

Specific comments

Methods - Page 3, row 117: "Day time emissions" could be expressed more specifically. In the previous sentence you say that total field measurement time varied from 4 to 10 hours. It is not totally clear, did you visit each lake ones, and spent in average 6 hours there? More useful would be to know what is "daytime measurement" 10-16 (6 hours)? 12-18 (six hours)? You could say e.g. all measurement were carried out between 9am and 6 pm. Or did you visit the same lake several times? - Page 3, row 117: I would put CH4 first, it is bit confusing that you mention CO2 before CH4, the whole manuscript is anyway about methane. You could also write here that

CO2 was measured as a background information. - Page 4, row 149: The whole idea of comparing West Siberia lakes with lakes in South Eastern Australia seems rather strange. Same "West Siberian climate is more similar to the Australian autumn." Do you mean West Siberian summers resemble Australian autumns? - Page 4, row 159: Do you mean 10 cm above sediment? - Page 4, rows 159-160: Please add here all the abbreviations you use later in the text. - Page 5, row 170: You could specify trace metals, the correlation between CH4 and Cu comes out of the blue for the reader later in the Results section, because Cu has not been mentioned previously. - Page 5, row 190: You could shortly explain here which are the "well-known controls". E.g. the first two sentences from Appendix A (Page 15, row 558-561) could be here instead of being in the Appendix. - Page 6, row 226: I would put this paragraph first or at least somewhere in the beginning of the section. Now there are lengthy explanations before you actually say the main point of the chapter.

Results - Page 6, row 243-244: You have not explained the abbreviations EC or Eh previously. - Page 7, row 266-267: I would delete the last sentence. I don't think it is necessary to tell were you are going to discuss the results. - Page 7, row 268: This is rather surprising considering that there are significant relationships in the simple regression analysis. I'm not familiar with Statistica, but the in the softwater I have used, multiple regression gives the similar result as the single one. If only one variable shows significant relationship, the multiple regression procedure includes only that variable. I think you could explain the multiple regression method in more detail.

Discussion - Page 8, row 321: According to the Table 2 there are considerable pH differences between MT and ST lake populations, with significantly higher pH in ST lakes. In the Figure B1 you present the optimum pH for the methane production. Although you say in the appendix that you have observed high methane production rates in the very low pH, the reader anyhow sees the Figure and the pH difference between lake populations. I think you should discuss the influence of pH differences between the lake populations in this section. - Page 8, row 330: add: Net primary production (NPP)

- Page 9, row 348: I'm wondering, if the both wetlands are acidic, why are there considerable pH differences between MT and ST lakes? - Page 10, row 379: This could be explained in more detail, since Cu was the only element correlating with CH4 in your data. I think you could already in the Introduction tell a little about what is known about controls and inhibitors of methanotrophy and mentanogenesis. - Page 12, row 497: What do you mean CO2 and CH4 fluxes almost the same? Methane fluxes are about 100 times higher according to Table 4? I think you mean that there is very little variation between MT lakes in both CO2 and CH4 fluxes?

- Table 2 does not include all the water quality variables that you measured. It would be nice to know how e.g. Cu varies, because it correlates with CH4.

- Table 6: There are empty rows in the first "Reference" column. What are the numbers presented in the next columns?

---

## Referee Comment (RC2) · Anonymous Referee #2 · 27 Oct 2016

**Comments to the Author,**

**Review of "Variability in methane emissions from West Siberia's shallow boreal lakes"**

**Summary**

The manuscript describes a study of CH4 emissions from boreal shallow lakes (14 lakes), with distinct limnological characteristics across two taiga zones in west Siberia. Authors used static chamber and bubble traps to estimate the total CH4 emissions. The aim of the study was to compare the magnitude and variability of CH4 emissions between lake at different zones, and among lakes. To achieve this, CH4 emissions and environmental controls were used in a new dynamic process-based model. The main idea to use this model is the fact that CH4 emissions are not predictable, uniform nor spatiotemporal distributed. Therefore, self-organized critically theory (model theorem) can help to assess the spatiotemporal heterogeneity of CH4 emissions.

The study of CH4 emissions from lakes is a topic of broad scientific interest as lakes represent important sources of this gas to the atmosphere. Moreover, nowadays lakes represent an important uncertain in the global CH4 budget and more information is needed to improve the current estimations. The value of this manuscript is that it shows an overview across lakes located in an area scarcely studied but with important quantity of lakes; jointly with the possibility to use dynamic process-based model to improve the knowledge of the CH4 cycling in lakes. Hence, the manuscript is a potential contribution to Biogeosciences. However, there are some aspects in the manuscript that could be improved to enhance the value of the acquired information.

My major concern is the idea of using the new dynamic process-based model to improve precision on model CH4 emission among biogeochemical attributes. Because, measurements of spatiotemporal CH4 emissions and biogeochemical parameters were scarcely done. As Patrick Crill pointed "data without models are chaos, but models without data are fantasy" (mentioned in Nisbet et al. 2014), therefore, poor measurements promote data inconsistency and inability to extrapolate estimations accurately. In this manuscript, the justification to use this model is very subjective, since some parameters were poorly measured and/or taken from literature (e.g. dissolved CH4 concentration in water surface, ebullition traps, physicochemical sediment information). I would ask them to present a better explanation for the use of that model and the scope of it. Because, as it stands, it makes me think that the lack of actual data collected from the field has influenced the poor performance for individual lakes in the middle taiga region.

**Specific comments**

**Introduction**

Page 1, row 37-41: New and important manuscripts had published recently about CH4 emissions of small ponds and boreal lakes, and lake distribution that can be included in the references: Holgerson and Raymond (2016), Wik et al. (2016b), Saunois et al. (2016) and Verpoorter et al. (2014). Besides, according to the new assessment of Saunois et al. (2016), lakes emit a range of 37 to 112 Tg CH4 per year; so, you can include this current estimation in the text.

Pages 1-2, row 41-44: It should be the first part of this paragraph to follow from general to specific ideas.

Page 2, row 44-46: Could you go deeply in this statement? I recommend Nisbet et al. (2014) and Saunois et al. (2016) literature to improve this idea.

Page 2, row 48: I suggest to include hot-topic references on this point, and even include temperature dependence on CH4 production in lake sediment assessments. For example: Schulz et al. (1997), Marotta et al. (2014), Yvon-Durocher et al. (2014). Maybe you can remove Kotsyurbenko et al (2001), since it is a study of reactor sludge and competition between methanogens and sulfate reducers bacteria.

Page 2, row 49-50: I can't find in Juutinen et al. (2009) manuscript this statement. They even pointed that CH4 oxidation was large in a shallow Lake Kevätön before spring-overturn (when they talk about their CH4 budget). Martinez-Cruz et al. (2015) found a very active methanotrophy in water column of shallow lakes from an Alaska North-South transect. From those lakes and others reported in Sepulveda-Jauregui et al. (2015), 10 shallow lakes presented stratification during summer. Which is a common pattern in ecosystems rich in DOC (see Williamnson et al. 1999).

Page 2, row 53-61: About spatial CH4 emission variability and factors that controls them. I recommend to check new and hot-topic literature. For example: Wik et al. (2016a), Schilder et al. (2016), DelSontro et al. (2011, 2015 and 2016), Hofman et al. (2013), West et al. (2015), Natchimuthu et al. (2015) among others.

Page 2, row 63: You need a better connection to link in previous paragraphs about CH4 dynamics in lakes and the regional study.

Page 2, row 82: I would recommend to include some studies previous mentioned from DelSontro's group since they have interesting approaches to study bubbling variability.

**Material and Methods**

Page 3, row 102-111:
Tables
Table 1. Could you give a range of these values instead means? Why data are from 1979 to 2007? Additionally, I think there are few information on this table, therefore, I recommend to include more climatic characteristics, or just mentioned it in the text and avoid a poor Table information.
Table 2. If you measure different sections and or sites, you can give a range and/or the variability in each data reported.
Text
How do you define humic lake? This information is missing here or in Table 2. Moreover, I cannot see how you determine trophic state mentioned for some lakes, moreover, in others lakes I don't have idea of the trophic state and the method used to determine it. Finally, sediment information needs to be acknowledged.

Page 3, row 116-117: What is the advantage to use a "rubber" boat to prevent any influence on the lake vegetation and sediment?

Page 3-4, row 120-132: How do you store the syringes? I mean there is a strong possibility of leaks and permeability with these syringes type. Did you have a control to check this problem? Additionally, how do you divide the syringes for CH4 and CO2 measurements? You may indicate the number of measurements per sampling point for each gas.

Page 4, row 133-135: It is very confusing to me, please organize the idea and include more information. For example, headspace volume and water volume, concentration of CH4 "known", where gas sample was stored.

Page 4, row 141-143: I am not convinced of this statement, since shallow lakes and ponds in boreal regions has been presented stratification (e.g. Bouchard et al. 2015, Sepulveda-Jauregui et al. 2015). Additionally, you need to discuss deeply about single daytime measurements and possible bias in the flux estimates. Bastviken's and co-researchers are working nicely in this topic (Wik et al. 2016a, Schilder et al. 2016, Peixoto et al. 2016, Natchimuthu et al. 2015, Natchimuthu et al. 2014, among others).

Page 4, row 148-151: I think, a cite is not reliable to support such statement of comparing between Australian with Siberian lakes. Moreover, you can't justify your statement of "no store flux in your lakes", when your study covers only single day measurement in summer. What happen in spring turnover? If you have humic lakes and well protected by forest, then they may present a stratification period in warm summers.

Page 4, row 159-161: Please check the sentence meaning.

Page 4, row 161-163: Which was the device used to collect the water samples?

Page 5, row 170: What were the trace metals measured?

Page 5, row 191: Figure 2 and Model Structure. Oxygen production in the model is not considered (A2 equation and discussed in Page 17, row 657-658), however, you could explain better the reason and avoid like this statement: "no data about solar radiation is available". Why is not important O2 in CH4 oxidation (aerobic I think)? Why is primary production minimal in this model?

Page 5, row 202-203: You didn't measure pH in sediments and therefore you are overinterpreting with the water pH results. Therefore, it could influence in the idea to use pH in the model. I pointed this because, sediments contains important quantities of pH regulators, so, pH in sediments is commonly higher than pH in the water column (in acidic lakes). For example, in studies of CH4 cycling in an acidic bog lake in Germany (divided in four sections), pH in sediments from the acidic section was ranged from 5.9 to 6.0 in the first 20 cm, while pH in the water column was ranged from 4.2 to 4.6 (Casper et al. 2003).

Page 5-6, row 208-2010: Maybe you can refine this values with the studies made in sediments by Flury et al. (2015).

**Results and Discussion**

Page 7, row 258: is sample size enough to use two-sample Kolmogorov-Smirnov test?

Page 7, row 292: Please, add the references.

Page 8, row 228-331: This statement is out of the scope since you didn't study plant productivity. NPP is not described in the text.

Page 9, row 337-343: Ebullition traps were used 80% in ST and 30% MT of the lakes, so, you need to acknowledge that your data contains important uncertainties. As mentioned above, please review Bastviken's research and DelSontro's research about the spatial variability and distribution of the ebullition (even you see Anthony et al. 2010, Anthony and Anthony et al. 2013). Are your traps enough to be representative of the CH4 ebullition pathway? Additionally, please indicate the similarity order between your ebullition data and Repo et al. (2007).

Page 9, row 444: This sentence is confuse, please rephrase it.

Page 10, row 384-409: Flury et al. (2015) study can enhance the idea in this discussion section.

Page 12, row 486: West et al. (2015) and DelSontro et al. (2016) studies can enhance the idea in this sentence.

**References**

Anthony KMW, Anthony P. 2013. Constraining spatial variability of methane ebullition seeps in thermokarst lakes using point process models. Journal of Geophysical Research-Biogeosciences 118:1015-1034.

Anthony KMW, Vas DA, Brosius L, Chapin FS, Zimov SA, Zhuang QL. 2010. Estimating methane emissions from northern lakes using ice-bubble surveys. Limnology and Oceanography-Methods 8:592-609.

Bouchard F, Laurion I, Prekienis V, Fortier D, Xu X, Whiticar MJ. 2015. Modern to millennium-old greenhouse gases emitted from ponds and lakes of the Eastern Canadian Arctic (Bylot Island, Nunavut). Biogeosciences 12:7279-7298.

Casper P, Chan OC, Furtado ALS, Adams DD. 2003. Methane in an acidic bog lake: The influence of peat in the catchment on the biogeochemistry of methane. Aquatic Sciences 65:36-46.

DelSontro T, Boutet L, St-Pierre A, del Giorgio PA, Prairie YT. 2016. Methane ebullition and diffusion from northern ponds and lakes regulated by the interaction between temperature and system productivity. Limnology and Oceanography:n/a-n/a.

DelSontro T, Kunz MJ, Kempter T, Wuest A, Wehrli B, Senn DB. 2011. Spatial Heterogeneity of Methane Ebullition in a Large Tropical Reservoir. Environmental Science & Technology 45:9866-9873.

DelSontro T, McGinnis DF, Wehrli B, Ostrovsky I. 2015. Size Does Matter: Importance of Large Bubbles and Small-Scale Hot Spots for Methane Transport. Environmental Science & Technology 49:1268-1276.

Flury S, Glud RN, Premke K, McGinnis DF. 2015. Effect of Sediment Gas Voids and Ebullition on Benthic Solute Exchange. Environmental Science and Technology 49:10413–10420.

Hofmann H. 2013. Spatiotemporal distribution patterns of dissolved methane in lakes: How accurate are the current estimations of the diffusive flux path? Geophysical Research Letters 40:2779-2784.

Holgerson MA, Raymond PA. 2016. Large contribution to inland water $CO_2$ and $CH_4$ emissions from very small ponds. Nature Geosci 9:222-226.

Lofton DD, Whalen SC, Hershey AE. 2014. Effect of temperature on methane dynamics and evaluation of methane oxidation kinetics in shallow Arctic Alaskan lakes. Hydrobiologia 721:209-222.

Marotta H, Pinho L, Gudasz C, Bastviken D, Tranvik LJ, Enrich-Prast A. 2014. Greenhouse gas production in low-latitude lake sediments responds strongly to warming. Nature Climate Change 4:467-470.

Martinez-Cruz K, Sepulveda-Jauregui A, Anthony KW, Thalasso F. 2015. Geographic and seasonal variation of dissolved methane and aerobic methane oxidation in Alaskan lakes. Biogeosciences 12:4595-4606.

Natchimuthu S, Selvam BP, Bastviken D. 2014. Influence of weather variables on methane and carbon dioxide flux from a shallow pond. Biogeochemistry 119:403-413.

Natchimuthu S, Sundgren I, Gålfalk M, Klemedtsson L, Crill P, Danielsson Å, Bastviken D. 2015. Spatio-temporal variability of lake CH4 fluxes and its influence on annual whole lake emission estimates. Limnology and Oceanography:n/a-n/a.

Nguyen TD, Crill P, Bastviken D. 2010. Implications of temperature and sediment characteristics on methane formation and oxidation in lake sediments. Biogeochemistry 100:185-196.

Nisbet EG, Dlugokencky EJ, Bousquet P. 2014. Methane on the Rise-Again. Science 343:493-495.

Saunois M, et al. 2016. The Global Methane Budget: 2000-2012. Earth Syst. Sci. Data Discuss. 2016:1-79.

Schilder J, Bastviken D, van Hardenbroek M, Heiri O. 2016. Spatiotemporal patterns in methane flux and gas transfer velocity at low wind speeds: Implications for upscaling studies on small lakes. Journal of Geophysical Research: Biogeosciences 121:1456-1467.

Schulz S, Matsuyama H, Conrad R. 1997. Temperature dependence of methane production from different precursors in a profundal sediment (Lake Constance). Fems Microbiology Ecology 22:207-213.

Sepulveda-Jauregui A, Anthony KMW, Martinez-Cruz K, Greene S, Thalasso F. 2015. Methane and carbon dioxide emissions from 40 lakes along a north-south latitudinal transect in Alaska. Biogeosciences 12:3197-3223.

Verpoorter C, Kutser T, Seekell DA, Tranvik LJ. 2014. A global inventory of lakes based on high-resolution satellite imagery. Geophysical Research Letters 41:6396-6402.

West WE, Creamer KP, Jones SE. 2015. Productivity and depth regulate lake contributions to atmospheric methane. Limnology and Oceanography:n/a-n/a.

Wik M, Thornton BF, Bastviken D, Uhlbaeck J, Crill PM. 2016a. Biased sampling of methane release from northern lakes: A problem for extrapolation. Geophysical Research Letters 43:1256-1262.

Wik M, Varner RK, Anthony KW, MacIntyre S, Bastviken D. 2016b. Climate-sensitive northern lakes and ponds are critical components of methane release. Nature Geoscience 9:99-+.

Williamson CE, Morris DP, Pace ML, Olson AG. 1999. Dissolved organic carbon and nutrients as regulators of lake ecosystems: Resurrection of a more integrated paradigm. Limnology and Oceanography 44:795-803.

Yvon-Durocher G, Allen AP, Bastviken D, Conrad R, Gudasz C, St-Pierre A, Thanh-Duc N, del Giorgio PA. 2014. Methane fluxes show consistent temperature dependence across microbial to ecosystem scales Nature 507:488-491.

---

## Author Comment (AC1) · 20 Feb 2017

Response to reviewers comments are given in the supplement

Please also note the supplement to this comment:
http://www.biogeosciences-discuss.net/bg-2016-331/bg-2016-331-AC1-supplement.pdf

---

## Author Comment (AC2) · 21 Feb 2017

Dear Prof Stoy

Please find our response to reviewers comments. We are grateful for the reviewers comments and we agree to substantial changes to the ms and the model in response to the valuable input from the two reviewers. Our response to the reviewer's comments are detailed point by point below.

Yours sincerely

Response to reviewers comments

RC1 Anonymous referee #1

<…>

*I also recommend the language check by the native English speaker.*

Thank you for this recommendation, which has prompted further thorough investigation of the text by the manuscript's one co-author who is also a native speaker of English. The text will be significantly reworked for clarity and grammatical precision in a number of places.

<…>

Specific comments

Methods

*- Page 3, row 117: "Day time emissions" could be expressed more specifically. In the previous sentence you say that total field measurement time varied from 4 to 10 hours. It is not totally clear, did you visit each lake ones, and spent in average 6 hours there? More useful would be to know what is "daytime measurement" 10-16 (6 hours)? 12-18 (six hours)? You could say e.g. all measurement were carried out between 9am and 6 pm. Or did you visit the same lake several times?*

No, we visited all lakes one time. We will clarify it as you have suggested.

*- Page 3, row 117: I would put CH4 first, it is bit confusing that you mention CO2 before CH4, the whole manuscript is anyway about methane. You could also write here that CO2 was measured as a background information.*

We will do it.

*- Page 4, row 149: The whole idea of comparing West Siberia lakes with lakes in South Eastern Australia seems rather strange. Same "West Siberian climate is more similar to the Australian autumn." Do you mean West Siberian summers resemble Australian autumns?*

Yes, we try to say, that weather conditions during summers in Western Siberia are the same as during Australian autumns. There are a lot of cool windy cloudy days and almost no calm days with zero cloudiness. We will try to clarify it in the paper text.

*- Page 4, row 159: Do you mean 10 cm above sediment?*

Yes, sorry for mistake.

*- Page 4, rows 159-160: Please add here all the abbreviations you use later in the text.*

We will do it.

*- Page 5, row 170: You could specify trace metals, the correlation between CH4 and Cu comes out of the blue for the reader later in the Results section, because Cu has not been mentioned previously.*

We will do it.

*- Page 5, row 190: You could shortly explain here which are the "well-known controls". E.g. the first two sentences from Appendix A (Page 15, row 558-561) could be here instead of being in the Appendix.*

We will add this information.
- *Page 6, row 226: I would put this paragraph first or at least somewhere in the beginning of the section. Now there are lengthy explanations before you actually say the main point of the chapter.*
This paragraph will be replaced.

Results
- *Page 6, row 243-244: You have not explained the abbreviations EC or Eh previously.*
We will fix it: EC is electrical conductivity, Eh is oxidation-reduction potential.

- *Page 7, row 266-267: I would delete the last sentence. I don't think it is necessary to tell were you are going to discuss the results.*
We will remove this sentence.

- *Page 7, row 268: This is rather surprising considering that there are significant relationships in the simple regression analysis. I'm not familiar with Statistica, but the in the softwater I have used, multiple regression gives the similar result as the single one. If only one variable shows significant relationship, the multiple regression procedure includes only that variable. I think you could explain the multiple regression method in more detail.*
We just wanted to say that we cannot find significant relationships with two or more independent variables. We will correct this sentence.

Discussion
- *Page 8, row 321: According to the Table 2 there are considerable pH differences between MT and ST lake populations, with significantly higher pH in ST lakes. In the Figure B1 you present the optimum pH for the methane production. Although you say in the appendix that you have observed high methane production rates in the very low pH, the reader anyhow sees the Figure and the pH difference between lake populations. I think you should discuss the influence of pH differences between the lake populations in this section.*
We agree with the referee, that we should discuss influence of pH differences between the lake populations more consistently. Strictly speaking, we have already pointed out that there are lakes with low pH in ST and for these lakes model shows relatively good performance. That's why poor model performance for MT lakes cannot be explained by pH differences between zones. There are some literature data that lakes with different pH have different pH optimums for methanogenesis (Dunfield et al., 1993; Segers, 1998), but it can explain not more than 20-30% difference and not 10 times, as we see in the model. Therefore we suppose that at the moment pH differences presented good enough in the model.

- *Page 8, row 330: add: Net primary production (NPP)*
We will fix it, thank you.

- *Page 9, row 348: I'm wondering, if the both wetlands are acidic, why are there considerable pH differences between MT and ST lakes?*
Only for these two wetlands data about MMPR are available. It doesn't reflect the general variability of pH values in wetlands of these regions. In general, MT lakes statistically are more acidic than ST lakes because in MT almost all lakes are linked to the wetlands and nearly all wetlands in MT are acidic. In ST variability of lake trophic state is much higher. Some ST lakes have ground water supply, and hence are less acidic. This high ST lakes trophic state variability (in comparison with MT) can be explained by more pronounced relief and higher level of nutrient supply.

*- Page 10, row 379: This could be explained in more detail, since Cu was the only element correlating with CH4 in your data. I think you could already in the Introduction tell a little about what is known about controls and inhibitors of methanotrophy and methanogenesis.*
We agree to add some information about elements, controlling main microbial processes of the methane cycle to the Introduction. Detailed explanation of possible Cu influence will be added too. Unfortunately, we don't have any data to check the importance of this explanation, but we can declare it as a hypothesis.

*- Page 12, row 497: What do you mean CO2 and CH4 fluxes almost the same? Methane fluxes are about 100 times higher according to Table 4? I think you mean that there is very little variation between MT lakes in both CO2 and CH4 fluxes?*
Yes, we just want to say that variation between MT lakes in both CO2 and CH4 fluxes is very little and not higher than errors of flux measurements. We will correct this phase.

*- Table 2 does not include all the water quality variables that you measured. It would be nice to know how e.g. Cu varies, because it correlates with CH4.*
We will add this information. It was not included only because resulting table was very big.

*- Table 6: There are empty rows in the first "Reference" column. What are the numbers presented in the next columns?*
It is because information about more than one lake was borrowed from several papers (namely Juutinen et al., 2009, Repo et al., 2007 and Bastviken et al., 2008). We will try to remove repeating climate information to show the reader that information in two or three rows was borrowed from one source.

RC1 Anonymous referee #2

Summary
<...>
*My major concern is the idea of using the new dynamic process-based model to improve precision on model CH4 emission among biogeochemical attributes. Because, measurements of spatiotemporal CH4 emissions and biogeochemical parameters were scarcely done. As Patrick Crill pointed "data without models are chaos, but models without data are fantasy" (mentioned in Nisbet et al. 2014), therefore, poor measurements promote data inconsistency and inability to extrapolate estimations accurately. In this manuscript, the justification to use this model is very subjective, since some parameters were poorly measured and/or taken from literature (e.g. dissolved CH4 concentration in water surface, ebullition traps, physicochemical sediment information). I would ask them to present a better explanation for the use of that model and the scope of it. Because, as it stands, it makes me think that the lack of actual data collected from the field has influenced the poor performance for individual lakes in the middle taiga region.*

We agree with referee, that obtained data have several gaps. But we think that these gaps should not lead to overinterpretation of the model. We believe that this model is not a fantasy because we try to rely on basic principles and well-known dependencies. Partly we try to check how we can explain methane variability using parameters from literature. We do not use any calibration or selection of parameters and use average values where it was possible. We think that this approach can show us possible gaps in our knowledge about methane cycle in shallow boreal lakes.

Specific comments
Introduction
*Page 1, row 37-41: New and important manuscripts had published recently about CH4 emissions of small ponds and boreal lakes, and lake distribution that can be included in the*

*references: Holgerson and Raymond (2016), Wik et al. (2016b), Saunois et al. (2016) and Verpoorter et al. (2014). Besides, according to the new assessment of Saunois et al. (2016), lakes emit a range of 37 to 112 Tg CH4 per year; so, you can include this current estimation in the text.*
Thank you for very useful references, we will use it in the paper text.

*Pages 1-2, row 41-44: It should be the first part of this paragraph to follow from general to specific ideas.*
We will replace it.

*Page 2, row 44-46: Could you go deeply in this statement? I recommend Nisbet et al. (2014) and Saunois et al. (2016) literature to improve this idea.*
We did not know about these new papers and will use these papers about lakes as hot-spots.

*Page 2, row 48: I suggest to include hot-topic references on this point, and even include temperature dependence on CH4 production in lake sediment assessments. For example: Schulz et al. (1997), Marotta et al. (2014), Yvon-Durocher et al. (2014). Maybe you can remove Kotsyurbenko et al (2001), since it is a study of reactor sludge and competition between methanogens and sulfate reducers bacteria.*
We will do it, thank you very much.

*Page 2, row 49-50: I can't find in Juutinen et al. (2009) manuscript this statement. They even pointed that CH4 oxidation was large in a shallow Lake Kevätön before spring overturn (when they talk about their CH4 budget). Martinez-Cruz et al. (2015) found a very active methanotrophy in water column of shallow lakes from an Alaska North-South transect. From those lakes and others reported in Sepulveda-Jauregui et al. (2015), 10 shallow lakes presented stratification during summer. Which is a common pattern in ecosystems rich in DOC (see Williamnson et al. 1999).*
We will change it. Probably we lost sense rewording the phrase.

*Page 2, row 53-61: About spatial CH4 emission variability and factors that controls them. I recommend to check new and hot-topic literature. For example: Wik et al. (2016a), Schilder et al. (2016), DelSontro et al. (2011, 2015 and 2016), Hofman et al. (2013), West et al. (2015), Natchimuthu et al. (2015) among others.*
Thank you for useful references, we will use it in the paper text.

*Page 2, row 63: You need a better connection to link in previous paragraphs about CH4 dynamics in lakes and the regional study.*
We will try to provide it and try to put more attention to fact that there is lack of papers about spatial methane flux differences on the regional scale. And it is especially interesting in Western Siberia, where almost whole region is covered by numerous lakes.

*Page 2, row 82: I would recommend to include some studies previous mentioned from DelSontro's group since they have interesting approaches to study bubbling variability.*
We will use it.

Material and Methods
Page 3, row 102-111:
Tables
*Table 1. Could you give a range of these values instead means?*
Yes, we will do it.

*Why data are from 1979 to 2007?*
We will add more fresh data.

*Additionally, I think there are few information on this table, therefore, I recommend to include more climatic characteristics, or just mentioned it in the text and avoid a poor Table information.*
We will mention this information in the text.

*Table 2. If you measure different sections and or sites, you can give a range and/or the variability in each data reported.*
We think that it would lead to the overload of the table and is not informative. But if referee thinks it would be better for paper, we will do it.

*Text*
*How do you define humic lake? This information is missing here or in Table 2. Moreover, I cannot see how you determine trophic state mentioned for some lakes, moreover, in others lakes I don't have idea of the trophic state and the method used to determine it. Finally, sediment information needs to be acknowledged.*
We determine trophic state according to (Wetzel, 2001) based on phosphorus and sulfate concentration. Sediment information will be added.

*Page 3, row 116-117: What is the advantage to use a "rubber" boat to prevent any influence on the lake vegetation and sediment?*
Of course, it is not important from what material boat was made. We will remove word "rubber".

*Page 3-4, row 120-132: How do you store the syringes? I mean there is a strong possibility of leaks and permeability with these syringes type. Did you have a control to check this problem?*
$CH_4$ samples stored in salt boiled water. $CO_2$ samples were analyzes in 1-4 hours after sampling, we did not store it. Before the beginning we checked the intensity of leakages. We will add this information.

*You may indicate the number of measurements per sampling point for each gas.*
We will add this information. We have 4 samples for each flux calculation, each sample was analyzed in three replicates.

*Page 4, row 133-135: It is very confusing to me, please organize the idea and include more information. For example, headspace volume and water volume, concentration of $CH_4$ "known", where gas sample was stored.*
We will do it.

*Page 4, row 141-143: I am not convinced of this statement, since shallow lakes and ponds in boreal regions has been presented stratification (e.g. Bouchard et al. 2015, Sepulveda-Jauregui et al. 2015). Additionally, you need to discuss deeply about single daytime measurements and possible bias in the flux estimates. Bastviken's and co-researchers are working nicely in this topic (Wik et al. 2016a, Schilder et al. 2016, Peixoto et al. 2016, Natchimuthu et al. 2015, Natchimuthu et al. 2014, among others).*
We will add this discussion.

*Page 4, row 148-151: I think, a cite is not reliable to support such statement of comparing between Australian with Siberian lakes. Moreover, you can't justify your statement of "no store flux in your lakes", when your study covers only single day measurement in summer. What*

*happen in spring turnover? If you have humic lakes and well protected by forest, then they may present a stratification period in warm summers.*
We agree and will rewrite these sentences.

*Page 4, row 159-161: Please check the sentence meaning.*
We think everything is correct except "10 cm below sediment depth". It should be "above".

*Page 4, row 161-163: Which was the device used to collect the water samples?*
Long plastic tube. We will add it.

*Page 5, row 170: What were the trace metals measured?*
We will add this information.

*Page 5, row 191: Figure 2 and Model Structure. Oxygen production in the model is not considered (A2 equation and discussed in Page 17, row 657-658), however, you could explain better the reason and avoid like this statement: "no data about solar radiation is available". Why is not important O2 in CH4 oxidation (aerobic I think)? Why is primary production minimal in this model?*
We will add these calculations to the model. We included simple model of oxygen production. After calculations we can say it does not change the result, because our shallow lakes were saturated with oxygen. But from general principles it was better to do.

*Page 5, row 202-203: You didn't measure pH in sediments and therefore you are overinterpreting with the water pH results. Therefore, it could influence in the idea to use pH in the model. I pointed this because, sediments contains important quantities of pH regulators, so, pH in sediments is commonly higher than pH in the water column (in acidic lakes). For example, in studies of CH4 cycling in an acidic bog lake in Germany (divided in four sections), pH in sediments from the acidic section was ranged from 5.9 to 6.0 in the first 20 cm, while pH in the water column was ranged from 4.2 to 4.6 (Casper et al. 2003).*
We will add new data about sediment water pH, obtained in 2015. They are sparse but it is better than nothing.

*Page 5-6, row 208-210: Maybe you can refine this values with the studies made in sediments by Flury et al. (2015).*
Thank you very much for this reference, it proofs previous estimates.

Results and Discussion
*Page 7, row 258: is sample size enough to use two-sample Kolmogorov-Smirnov test?*
We don't find a limit for this test.

*Page 7, row 292: Please, add the references.*
All references are given further in this paragraph after row 292. First sentence just announces it.

*Page 8, row 328-331: This statement is out of the scope since you didn't study plant productivity. NPP is not described in the text.*
Yes, but we don't use it in the paper for any calculations, it is a qualitative statement. We just wanted to proof our idea with this reference.

*Page 9, row 337-343: Ebullition traps were used 80% in ST and 30% MT of the lakes, so, you need to acknowledge that your data contains important uncertainties. As mentioned above, please review Bastviken's research and DelSontro's research about the spatial variability and distribution of the ebullition (even you see Anthony et al. 2010, Anthony and Anthony et al.*

*2013). Are your traps enough to be representative of the CH4 ebullition pathway? Additionally, please indicate the similarity order between your ebullition data and Repo et al. (2007).*
We will do it.

*Page 9, row 444: This sentence is confuse, please rephrase it.*
We will do it.

*Page 10, row 384-409: Flury et al. (2015) study can enhance the idea in this discussion section.*
*Page 12, row 486: West et al. (2015) and DelSontro et al. (2016) studies can enhance the idea in this sentence.*
We will try to include these high-level ideas to the paper text.

---

## Author Response (AR1)

Dear Prof. Stoy and referees,

Please find our response to referees comments. We are grateful for the reviewers comments and we agree to changes to the MS and the model in response to the valuable input from the two reviewers. Our response to the reviewer's comments are detailed point by point below.

Yours sincerely

Response to reviewers comments

RC1 Anonymous referee #1

<…>

*I also recommend the language check by the native English speaker.*

Thank you for this recommendation, which has prompted further thorough investigation of the text by the manuscript's one co-author who is also a native speaker of English.

<…>

Specific comments

Methods

*- Page 3, row 117: "Day time emissions" could be expressed more specifically. In the previous sentence you say that total field measurement time varied from 4 to 10 hours. It is not totally clear, did you visit each lake ones, and spent in average 6 hours there? More useful would be to know what is "daytime measurement" 10-16 (6 hours)? 12-18 (six hours)? You could say e.g. all measurement were carried out between 9am and 6 pm. Or did you visit the same lake several times?*

No, we visited each lake one time. We clarified it as you have suggested.

*- Page 3, row 117: I would put CH4 first, it is bit confusing that you mention CO2 before CH4, the whole manuscript is anyway about methane. You could also write here that CO2 was measured as a background information.*

Done.

*- Page 4, row 149: The whole idea of comparing West Siberia lakes with lakes in South Eastern Australia seems rather strange. Same "West Siberian climate is more similar to the Australian autumn." Do you mean West Siberian summers resemble Australian autumns?*

We try to say, that weather conditions during summers in Western Siberia are the same as during Australian autumns. There are a lot of cool windy cloudy days and almost no calm days with zero cloudiness. We will try to clarify it in the paper text.

We removed phrases about comparison. Now we compare situations up to the moment – during our measurements there was no stratification. This is what we wanted to say.

*- Page 4, row 159: Do you mean 10 cm above sediment?*

Yes, sorry for mistake.

*- Page 4, rows 159-160: Please add here all the abbreviations you use later in the text.*

Done.

*- Page 5, row 170: You could specify trace metals, the correlation between CH4 and Cu comes out of the blue for the reader later in the Results section, because Cu has not been mentioned previously.*

Done.

*- Page 5, row 190: You could shortly explain here which are the "well-known controls". E.g. the first two sentences from Appendix A (Page 15, row 558-561) could be here instead of being in the Appendix.*
Done.

*- Page 6, row 226: I would put this paragraph first or at least somewhere in the beginning of the section. Now there are lengthy explanations before you actually say the main point of the chapter.*
Done.

Results
*- Page 6, row 243-244: You have not explained the abbreviations EC or Eh previously.*
We fixed it: EC is electrical conductivity, Eh is oxidation-reduction potential.

*- Page 7, row 266-267: I would delete the last sentence. I don't think it is necessary to tell were you are going to discuss the results.*
Done.

*- Page 7, row 268: This is rather surprising considering that there are significant relationships in the simple regression analysis. I'm not familiar with Statistica, but the in the softwater I have used, multiple regression gives the similar result as the single one. If only one variable shows significant relationship, the multiple regression procedure includes only that variable. I think you could explain the multiple regression method in more detail.*
We just wanted to say that we cannot find significant relationships with two or more independent variables. It was fixed.

Discussion
*- Page 8, row 321: According to the Table 2 there are considerable pH differences between MT and ST lake populations, with significantly higher pH in ST lakes. In the Figure B1 you present the optimum pH for the methane production. Although you say in the appendix that you have observed high methane production rates in the very low pH, the reader anyhow sees the Figure and the pH difference between lake populations. I think you should discuss the influence of pH differences between the lake populations in this section.*
Strictly speaking, we have already pointed out that there are lakes with low pH in ST sample and for these lakes model shows relatively good performance. That's why poor model performance for MT lakes cannot be explained by pH differences between zones. There are some literature data that lakes with different pH have different pH optimums for methanogenesis (Dunfield et al., 1993; Segers, 1998), but it can explain not more than 20-30% difference and not 10 times, as we see in the model. Therefore we suppose that at the moment pH differences presented good enough in the model.

*- Page 8, row 330: add: Net primary production (NPP)*
Done, thank you.

*- Page 9, row 348: I'm wondering, if the both wetlands are acidic, why are there considerable pH differences between MT and ST lakes?*
Only for these two acidic wetlands data about MMPR are available. It doesn't reflect the general variability of pH values in lakes of these regions because not each lake has wetlands nearby. In general, MT lakes statistically are more acidic than ST lakes because in MT almost all lakes are linked to the wetlands and nearly all wetlands in MT are acidic. In ST variability of lake trophic state is much higher. Some ST lakes have ground water supply, and hence are less acidic. This

high ST lakes trophic state variability (in comparison with MT) can be explained by more pronounced relief and higher level of nutrient supply.

*- Page 10, row 379: This could be explained in more detail, since Cu was the only element correlating with CH4 in your data. I think you could already in the Introduction tell a little about what is known about controls and inhibitors of methanotrophy and methanogenesis.*
In the introduction we prefer just to declare controls with number of references. Telling about each possible control in the Introduction unreasonably enlarges a paper. There are number of reviews with detailed description of methane emission environmental controls. Our paper seems to be too long already.
Discussion of possible Cu influence was added to the paper text. There was no enough data for that but we tried to give objective presentation.

*- Page 12, row 497: What do you mean CO2 and CH4 fluxes almost the same? Methane fluxes are about 100 times higher according to Table 4? I think you mean that there is very little variation between MT lakes in both CO2 and CH4 fluxes?*
Yes, we just want to say that variation between MT lakes in both CO2 and CH4 fluxes is very little and not higher than errors of flux measurements. We fixed this phase.

*- Table 2 does not include all the water quality variables that you measured. It would be nice to know how e.g. Cu varies, because it correlates with CH4.*
Done. It was not included only because resulting table was very big.

*- Table 6: There are empty rows in the first "Reference" column. What are the numbers presented in the next columns?*
These are numbers for each of several lakes or group of lakes. For several papers information about more than one lake was presented (namely Juutinen et al., 2009, Repo et al., 2007 and Bastviken et al., 2008). We will try to remove repeating climate information to show the reader that information in two or three rows was borrowed from one source.

RC1 Anonymous referee #2

Summary
<...>
*My major concern is the idea of using the new dynamic process-based model to improve precision on model CH4 emission among biogeochemical attributes. Because, measurements of spatiotemporal CH4 emissions and biogeochemical parameters were scarcely done. As Patrick Crill pointed "data without models are chaos, but models without data are fantasy" (mentioned in Nisbet et al. 2014), therefore, poor measurements promote data inconsistency and inability to extrapolate estimations accurately. In this manuscript, the justification to use this model is very subjective, since some parameters were poorly measured and/or taken from literature (e.g. dissolved CH4 concentration in water surface, ebullition traps, physicochemical sediment information). I would ask them to present a better explanation for the use of that model and the scope of it. Because, as it stands, it makes me think that the lack of actual data collected from the field has influenced the poor performance for individual lakes in the middle taiga region.*
We agree with referee, that obtained data have several gaps. But we think that these gaps should not lead to overinterpretation of the model. We believe that this model is not a fantasy because we try to rely on basic principles and well-known dependencies. Partly we try to check how we can explain methane variability using parameters from literature. We do not use any calibration or selection of parameters and use average values where it was possible. We think that this approach can show us possible gaps in our knowledge about methane cycle in shallow boreal

lakes. We added detailed description of the study scope in Methods section. Hope it removed a risks of overinterpretation.

Specific comments
Introduction
*Page 1, row 37-41: New and important manuscripts had published recently about CH4 emissions of small ponds and boreal lakes, and lake distribution that can be included in the references: Holgerson and Raymond (2016), Wik et al. (2016b), Saunois et al. (2016) and Verpoorter et al. (2014). Besides, according to the new assessment of Saunois et al. (2016), lakes emit a range of 37 to 112 Tg CH4 per year; so, you can include this current estimation in the text.*
Thank you for very useful references, we introduced them in the paper text.

*Pages 1-2, row 41-44: It should be the first part of this paragraph to follow from general to specific ideas.*
Fixed.

*Page 2, row 44-46: Could you go deeply in this statement? I recommend Nisbet et al. (2014) and Saunois et al. (2016) literature to improve this idea.*
Done.

*Page 2, row 48: I suggest to include hot-topic references on this point, and even include temperature dependence on CH4 production in lake sediment assessments. For example: Schulz et al. (1997), Marotta et al. (2014), Yvon-Durocher et al. (2014). Maybe you can remove Kotsyurbenko et al (2001), since it is a study of reactor sludge and competition between methanogens and sulfate reducers bacteria.*
Done.

*Page 2, row 49-50: I can't find in Juutinen et al. (2009) manuscript this statement. They even pointed that CH4 oxidation was large in a shallow Lake Kevätön before spring overturn (when they talk about their CH4 budget). Martinez-Cruz et al. (2015) found a very active methanotrophy in water column of shallow lakes from an Alaska North-South transect. From those lakes and others reported in Sepulveda-Jauregui et al. (2015), 10 shallow lakes presented stratification during summer. Which is a common pattern in ecosystems rich in DOC (see Williamnson et al. 1999).*
Fixed. Probably we lost sense rewording the phrase.

*Page 2, row 53-61: About spatial CH4 emission variability and factors that controls them. I recommend to check new and hot-topic literature. For example: Wik et al. (2016a), Schilder et al. (2016), DelSontro et al. (2011, 2015 and 2016), Hofman et al. (2013), West et al. (2015), Natchimuthu et al. (2015) among others.*
Done.

*Page 2, row 63: You need a better connection to link in previous paragraphs about CH4 dynamics in lakes and the regional study.*
Done as much as we can.

*Page 2, row 82: I would recommend to include some studies previous mentioned from DelSontro's group since they have interesting approaches to study bubbling variability.*
We think that we don't have enough data for this. We use bubble information in our paper as supporting information because these data are sparse and insufficient.

Material and Methods
Page 3, row 102-111:
Tables
*Table 1. Could you give a range of these values instead means?*
Done.

*Why data are from 1979 to 2007?*
Because we could not find better data. Now we used 1979-2014. We think, they present actual climate.

*Additionally, I think there are few information on this table, therefore, I recommend to include more climatic characteristics, or just mentioned it in the text and avoid a poor Table information.*
Done.

*Table 2. If you measure different sections and or sites, you can give a range and/or the variability in each data reported.*
We think that it would lead to the overload of the table and is not informative. But if referee thinks it would be better for paper, we will do it.

*Text*
*How do you define humic lake? This information is missing here or in Table 2. Moreover, I cannot see how you determine trophic state mentioned for some lakes, moreover, in others lakes I don't have idea of the trophic state and the method used to determine it.*
We removed this word, it is not a principal characteristic.

*Page 3, row 116-117: What is the advantage to use a "rubber" boat to prevent any influence on the lake vegetation and sediment?*
Of course, it is not important from what material boat was made. We removed word "rubber".

*Page 3-4, row 120-132: How do you store the syringes? I mean there is a strong possibility of leaks and permeability with these syringes type. Did you have a control to check this problem?*
$CH_4$ samples stored in salt boiled water. $CO_2$ samples were analyzes in 1-4 hours after sampling, we did not store it. Before the beginning we checked the intensity of leakages. We have added this information.

*You may indicate the number of measurements per sampling point for each gas.*
We have added this information. We have 4 samples for each flux calculation, each sample was analyzed in three replicates.

*Page 4, row 133-135: It is very confusing to me, please organize the idea and include more information. For example, headspace volume and water volume, concentration of CH4 "known", where gas sample was stored.*
Done.

*Page 4, row 141-143: I am not convinced of this statement, since shallow lakes and ponds in boreal regions has been presented stratification (e.g. Bouchard et al. 2015, Sepulveda-Jauregui et al. 2015).*
We agree that shallow lakes and ponds in boreal regions CAN have stratification. We have not written that our lakes NEVER have stratification. We just want to point out that during our measurements we do not see stratification (no strong temperature gradients higher than 1-2 degrees) and stratification does not bias our flux measurements.

*Additionally, you need to discuss deeply about single daytime measurements and possible bias in the flux estimates. Bastviken's and co-researchers are working nicely in this topic (Wik et al. 2016a, Schilder et al. 2016, Peixoto et al. 2016, Natchimuthu et al. 2015, Natchimuthu et al. 2014, among others).*

We did not pretend to present whole lake budgets or seasonal flux estimates in our paper. Of course single daytime session of measurements will not cover rare events of extreme gas bubbling. But with a help of presented probability distribution functions we tried to show that number of conducted measurement at each lake (average for our study is 14) is enough to present even very high fluxes (about 20 mgCH$_4$ m$^{-2}$ h$^{-1}$). That's why we can suggest, that in first approximation we reflect the methane emission up to the moment. Because our main target was spatial variability we construct the model for actual level of emission to show what factors are of special importance on this scale. By the way, Wik et al. 2016a suggests that 10-12 locations is enough to obtain high-accuracy CH$_4$ flux estimates. We have 14 flux replicates as average in our study for each lake. Chambers float in area of 5-10 m$^2$, all supporting data were obtained exactly for this area, and thus we can assume that actual level of emission for this point was successfully presented.

We have added the following discussion of possible bias to the paper text: It is important to notice that obtained flux and supporting data gives only an actual snapshot of methane emission from a certain lake section. Considering the spatio-temporal variability in CH4 fluxes is critical when making whole lake or annual budgets (Natchimuthu et al., 2015; Wik et al., 2016a) while our target is actual variability on a regional spatial scale. As it was mentioned above, the best way to take into account complicated and non-linear key processes is to construct a process-based model. Modern lake methane emission models tend to operate not with whole lake budgets but on a much smaller scale with subsequent averaging (Stepanenko et al., 2011, 2016; Tan et al., 2015) because it allows to resolve small-scale heterogeneity of such important controls as lake depth or water temperature. Considering not the whole lake but the certain lake sections can improve aquatic greenhouse gas emission estimates (Shilder et al., 2016). Therefore it should be mentioned that in our study not the whole lake but lake section (area about 10 m2) is the studied object.

*Page 4, row 148-151: I think, a cite is not reliable to support such statement of comparing between Australian with Siberian lakes. Moreover, you can't justify your statement of "no store flux in your lakes", when your study covers only single day measurement in summer. What happen in spring turnover? If you have humic lakes and well protected by forest, then they may present a stratification period in warm summers.*

We do not compare lakes, we compare situations in a certain period of the season. Again we speak only about period of our measurements; whole season dynamic is not the target. That's why we neglected storage flux but just in case of our measurements because we do not see any stratification. Of course there is a possibility of stratification and spring turnover but not during our summer field campaign. That's why we highlighted in the introduction that spatial variability on all scales is our target.

We also removed phrases in rows 148-151 because they are not necessary for the text.

*Page 4, row 159-161: Please check the sentence meaning.*

We think everything is correct except "10 cm below sediment depth". It should be "above".

*Page 4, row 161-163: Which was the device used to collect the water samples?*

Long plastic tube. We have added it.

*Page 5, row 170: What were the trace metals measured?*

We have added this information.

*Page 5, row 191: Figure 2 and Model Structure. Oxygen production in the model is not considered (A2 equation and discussed in Page 17, row 657-658), however, you could explain better the reason and avoid like this statement: "no data about solar radiation is available". Why is not important O2 in CH4 oxidation (aerobic I think)? Why is primary production minimal in this model?*

We have added these calculations to the model. We included simple model of oxygen production. After calculations we can say it does not change the result, because our shallow lakes were saturated with oxygen. But from general principles it was better to do.

*Page 5, row 202-203: You didn't measure pH in sediments and therefore you are overinterpreting with the water pH results. Therefore, it could influence in the idea to use pH in the model. I pointed this because, sediments contains important quantities of pH regulators, so, pH in sediments is commonly higher than pH in the water column (in acidic lakes). For example, in studies of CH4 cycling in an acidic bog lake in Germany (divided in four sections), pH in sediments from the acidic section was ranged from 5.9 to 6.0 in the first 20 cm, while pH in the water column was ranged from 4.2 to 4.6 (Casper et al. 2003).*

Lake from (Casper et al., 2003) study is NOT totally surrounded by huge acidic wetlands several meters in depth (as our sites in Western Siberia) and is fed by both ground water and precipitation (in Western Siberia – only atmospheric supply). That's why we do not think it is a good example for comparison.

Strictly speaking, there are no published data about pH in lake sediments for our sites or WS lakes. But there are pH profile data for surrounding wetlands. For MT there are data from (Sabrekov et al., 2011) where pH was measured in wetlands near lakes Lebedinoe and Babochka. This pH is 3.9-4.1, very close to lake pH in Lebedinoe and Babochka (4.2-4.4). For ST there are data from (Kotsyurbenko et al., 2004; 2007) where pH was measured near two studied acidic lakes Bakchar.bog.1-2; this pH is 4.8, very close to lake pH in lakes Bakchar.bog.1-2 (4.5-4.8). All studied acidic lakes have secondary origin (Kulkov et al., 2017), they have the same oligotrophic sphagnum peat in the bottom (at least – several upper meters which are only important for lake emission) as peat in wetlands around. This peat has very low ash content (Turunen et al., 2001) and unlikely able to strongly regulate pH.

Nevertheless, we agree, that we do not have data about pH lake sediment profile, and discussed it in the paper text.

List of references for these points:

Kotsyurbenko, O. R., Chin, K.-J., Glagolev, M. V., Stubner, S., Simankova, M. V., Nozhevnikova, A. N., and Conrad, R.: Acetoclastic and hydrogenotrophic methane production and methanogenic populations in an acidic West-Siberian peat bog, Environ. Microbiol., 6, 1159–1173, doi:10.1111/j.1462-2920.2004.00634.x, 2004.

Kotsyurbenko, O. R., Friedrich, M. W., Simankova, M. V., Nozhevnikova, A. N., Golyshin, P. N., Timmis, K. N., and Conrad, R.: Shift from acetoclastic to H2-dependent methanogenesis in a West Siberian peat bog at low pH values and isolation of an acidophilic *Methanobacterium* strain, Applied and Environmental Microbiology, 73(7), 2344–2348, doi: 10.1128/AEM.02413-06, 2007.

Kul'kov, M. G., Zarov, E. A., & Filippov, I. V. (2017). The choice of oil-pollution criteria for organogenic bottom sediments by chromatography-mass-spectrometry. Water Resources, 2(44), 267-275.

Sabrekov, A. F., Kleptsova, I. E., Glagolev, M. V., Maksyutov, S. S., and Machida, T.: Methane emission from middle taiga oligotrophic hollows of Western Siberia, Tomsk State Pedagogical University Bulletin, 5, 135–143, 2011. Available at: https://cyberleninka.ru/article/n/emissiya-metana-iz-oligotrofnyh-mochazhin-sredney-taygi-zapadnoy-sibiri (last access: 21 June 2017).

Turunen, J., Tahvanainen, T., Tolonen, K., and Pitkänen, A.: Carbon accumulation in West Siberian mires, Russia Sphagnum peatland distribution in North America and Eurasia during the past 21,000 years, Global Biogeochemical Cycles, 15(2), 285–296, doi:10.1029/2000GB001312, 2001.

*Page 5-6, row 208-210: Maybe you can refine this values with the studies made in sediments by Flury et al. (2015).*
Thank you very much for this reference, it proofs previous estimates.

*Results and Discussion*
*Page 7, row 258: is sample size enough to use two-sample Kolmogorov-Smirnov test?*
We did't find a limit for this test.

*Page 7, row 292: Please, add the references.*
All references are given further in this paragraph after row 292. First sentence just announces it.

*Page 8, row 328-331: This statement is out of the scope since you didn't study plant productivity. NPP is not described in the text.*
We don't use it in the paper for any calculations; it is a qualitative statement to show difference in NPP between zones using literature data. We just wanted to proof our idea with this reference.

*Page 9, row 337-343: Ebullition traps were used 80% in ST and 30% MT of the lakes, so, you need to acknowledge that your data contains important uncertainties. As mentioned above, please review Bastviken's research and DelSontro's research about the spatial variability and distribution of the ebullition (even you see Anthony et al. 2010, Anthony and Anthony et al. 2013). Are your traps enough to be representative of the CH4 ebullition pathway?*
Of course, representability of these data is not high, that's why we do not use it for any calculations. These data used only for comparison in a first approximation. Uncertainties were acknowledged in a paper text.

*Additionally, please indicate the similarity order between your ebullition data and Repo et al. (2007).*
Done.

*Page 9, row 444: This sentence is confuse, please rephrase it.*
Done.

*Page 10, row 384-409: Flury et al. (2015) study can enhance the idea in this discussion section.*
We have tried to introduce ideas from this paper but more for illustration, because this paper only suggests the hypothesis.

*Page 12, row 486: West et al. (2015) and DelSontro et al. (2016) studies can enhance the idea in this sentence.*
Thank you, it is very good for us that our hypothesis has proofs in recent publications. We included them to the paper text.

---

## Author Response (AR2)

Dear Prof. Stoy,

We are grateful for your estimation of our paper and hope that some our ideas will help the scientific community to improve its understanding of methane cycle in lakes, especially small boreal lakes. In this round of corrections we fixed several grammar mistakes, corrected references and figure 2 (introducing Photosynthesis into water column as was described in a paper text) and tried to clarify parts of each section of the document through changes in sentence structure and word choice. In response to your suggestion:

*I question the distinction of 'strong' and 'faint' correlations based on R2 thresholds as these distinctions seem to be chosen at random and one would expect few 'strong' (R2>0.5) relationships in the case of measuring multiple small lakes.*

We do not want to overrate our findings in these points. The initial idea was to point out that the correlations we found are not very clear and pronounced. Of course, this classification is not objective. So we removed these categories from the paper text. Actually, as all the necessary information about the correlations is given in the respective brackets, readers can see for themselves how strong or weak each correlation is. Thank you for this suggestion.

Yours sincerely